# Heavy mineral assemblages of the De Long Trough and southern Lomonosov Ridge glacigenic deposits: implications for the East Siberian Ice Sheet extent

Raisa Alatarvas[1], Matt O'Regan[2], Kari Strand[1]

[1]Oulu Mining School, University of Oulu, Oulu, 90570, Finland

[2]Department of Geological Sciences, Stockholm University, Stockholm, 106 91, Sweden

*Correspondence to*: Raisa Alatarvas (raisa.alatarvas@oulu.fi)

**Abstract.** The Arctic's glacial history has classically been interpreted from marine records in terms of the fluctuations of the Eurasian and North American ice sheets. However, the extent and timing of the East Siberian Ice Sheet (ESIS) have remained uncertain. A recently discovered glacially scoured cross-shelf trough extending to the edge of the continental shelf north of the De Long Islands has provided additional evidence that glacial ice existed on parts of the East Siberian Sea (ESS) during previous glacial periods MIS 6 and 4. This study concentrates on defining the heavy mineral signature of glacigenic deposits from the East Siberian continental margin which were collected during the 2014 SWERUS-C3 expedition. The cores studied are 20-GC1 from the East Siberian shelf, 23-GC1 and 24-GC1 from the De Long Trough (DLT), and 29-GC1 from the southern Lomonosov Ridge (LR). Heavy mineral assemblages were used to identify prominent parent rocks in hinterland and other sediment source areas. The parent rock areas include major eastern Siberian geological provinces such as the Omolon massif, the Chukotka Fold Belt, the Verkhoyansk Fold Belt, and possibly the Okhotsk–Chukotka Volcanic Belt. The primary riverine sources for the ESS sediments are the Indigirka and Kolyma rivers, which material was glacially eroded and re-deposited in the DLT. The higher abundances of amphiboles in the heavy mineral assemblages may indicate ESS paleovalley of the Indigirka River as a major pathway of sediments, while the Kolyma River paleovalley pathway relates to a higher share of pyroxenes and epidote. Mineralogical signature in the DLT diamicts, consisting predominantly of amphiboles and pyroxenes with minor content of garnet and epidote, show clear delivery from the eastern part of the ESIS. Although the physical properties of the DLT glacial diamict closely resembles a pervasive diamict unit recovered from the southern LR, their source material is slightly different. The assemblages with elevated amphibole and garnet content, along with higher titanite and ilmenite content of the southern LR ice-rafted diamict emphasise the Verkhoyansk Fold Belt as a possible primary source. The presence of glacial sediments and the recovered glacial-tectonic features on the East Siberian continental shelf and slope, along with the results from this heavy mineral analysis implicate that glacial ice not only grew out from the East Siberian shelf but also from the De Long Islands, and that there was also ice rafting related sediment transportation to the southern LR from westerly sources, such as the Laptev Sea.

## 1 Introduction

The existence, timing and extent of ice sheets during the Pleistocene glaciations on the Siberian continental shelf of the East Siberian Sea is still largely undetermined. The sparse geophysical and marine geological data obtained from this area are highly important in defining the glacial history of the region and its relationship to the wider Eurasian Arctic glaciations. Seafloor mapping data provide ample evidence for the existence of considerable ice masses on the East Siberian margin (Niessen et al., 2013; Jakobsson et al., 2014, 2016), but the timing and extent of these glaciations is still relatively unknown. According to West et al. (2021), there is a broad consensus on the lack of glacial activity on the Siberian shelf during the last glacial maximum (LGM), but when, and to what extent, the former ice sheets existed on the Siberian shelf remains poorly constrained by terrestrial evidence. This study concentrates on defining the heavy mineral signatureof marine glacial sediments from the

East Siberian continental margin, especially from a glacial trough and related trough mouth fan setting. Information about the glacial history of a continental margin can be obtained from the temporal evolution of trough mouth fan system (Stein, 2008). Studied sediment cores were collected during the SWERUS-C3 2014 expedition (SWERUS-C3: Swedish – Russian – US Arctic Ocean Investigation of Climate-Cryosphere-Carbon Interactions) from the East Siberian continental shelf and slope. The overreaching aim is to evaluate the heavy mineral compositions to see if there is a unique mineralogical signature in the diamict samples from the newly discovered DLT that can be used for far field reconstructions of ice sheet activity on the East Siberian continental margin. The mineralogical components and mineral-geochemical characteristics of these samples provide interpretative data for reconstruction of potential source areas and pathways for glacially entrained sediments. The relevance of this work is the utilisation of heavy mineral assemblages of the East Siberian shelf and slope sediments as provenance tracers. Examination of heavy minerals in the coarse fraction can be used in estimation of provenance and source areas, and plausible transport mechanisms. Most parent rock types, and their component minerals are presumably represented in the sand fraction in the straightforward case of subglacial erosion, transport, and deposition (Licht and Hemming, 2017).

## 2 Regional and geological setting

### 2.1 The East Siberian Sea and Shelf

The East Siberian Arctic Shelf extends ~2,500 km from the eastern Taimyr Peninsula coast from the East Siberian Sea (ESS) and Laptev Sea (LS) to the Russia and US marine border in the Chukchi Sea, containing the New Siberian and De Long Archipelagos Island groups, and Wrangel Island (Drachev et al., 2018). The East Siberian continental shelf extends over the North American Plate and the East Siberian Platform, and it is covered by the ESS and the LS. The shelf is composed of younger crust of the Late Mesozoic fold belts overlain with Late Mesozoic and Cenozoic siliciclastic sediments (Drachev et al., 2010). The East Siberian Sea Basin is filled with siliciclastic sediments with inferred stratigarphic range of Late Cretaceous to Quaternary age (Drachev et al., 2010).

The ESS is a relatively shallow sea. In the course of the larger glaciations following the mid-Pleistocene transition after 1.5 Ma, and during the sea level low stand of the Last Glacial Maximum (LGM), the region was most likely exposed due to most of the region having a depth of <120 m (Lambeck et al., 2014; Rohling et al., 2014). According to Klemann et al. (2015), throughout glacial periods, the ESS was largely exposed due to the global mean sea level being more than 100 m lower than its present value. During regressive and transgressive cycles, the shallowness of the shelf also might have led to the erosion of submarine glacial landforms which are an indication of the presence of ice sheets (Dowdeswell et al., 2016; O'Regan et al., 2017).

In the east, the continental slope of the ESS connects to the Mendeleev Ridge and Makarov Basin (O'Regan et al., 2016), with the latter adjoining the LR by extended continental crust (Jokat and Ickrath, 2015). The LR is an underwater ridge of continental crust expanding from the northern Greenland shelf, through the North Pole, to the central Siberian continental shelf, dividing the Arctic into the Eurasian and Amerasian basins (e.g., Kristoffersen et al., 1990; Cochran et al., 2006).

### 2.2 Glacial cross-shelf troughs and trough mouth fans

The existence of glacially excavated cross-shelf troughs (CSTs) indicate fast-streaming glacial ice areas on formerly glaciated margins (Batchelor and Dowdeswell, 2014). Bathymetrically prominent trough mouth fans (TMFs) are formed from large volumes of subglacial sediment discharging seaward of the shelf break onto the slope in front of CSTs (Ó Cofaigh et al., 2003; Batchelor and Dowdeswell, 2014). TMFs are stacked glacigenic debris flows interbedded with open-water or ice-distal marine sediments formed by fast-streaming ice transporting large volumes of subglacial sediments to the shelf break (Laberg and

Vorren, 1995; Elverhøi et al., 1997; Dowdeswell and Ó Cofaigh, 2002; Batchelor and Dowdeswell, 2014). Ó Cofaigh et al. (2003) concluded that the ideal conditions for TMF formation include a tectonically passive marginal setting; wide continental shelf; deformable, readily erodible sediments on the adjacent continental shelf; high rates of sediment delivery to the shelf edge; and, a low (<1°) slope gradient. According to Batchelor and Dowdeswell (2014) the formation of TMFs is more prominent in front of CSTs with recurrent ice stream activity through several glacial cycles, and where sediment is transported to comparatively shallow continental slopes.

Seismic and bathymetric data illustrate many glacially excavated troughs which discharged ice into the Arctic Ocean (Fig. 1) (Batchelor and Dowdeswell, 2014). Many of these troughs trace back in the direction of the center of former ice sheets, or back into branching fjords on adjoining landmasses (Batchelor and Dowdeswell, 2014; Jakobsson et al., 2016). Prior to the SWERUS-C3 2014 expedition (e.g. Jakobsson et al., 2016; O'Regan et al., 2017), no evidence for CSTs existed on the shallow shelf of the ESS. The DLT is the first glacial trough reported on the outer margin of the East Siberian shelf and is located north from the De Long and New Siberian Islands (Fig. 1) (O'Regan et al. 2017). The area interpreted as a TMF in front of the DLT totals 6540 km$^2$, with an average slope angle of 1.2 and thickness of the glacio-genic debris flow sequence of more than 65 m (O'Regan et al. 2017).

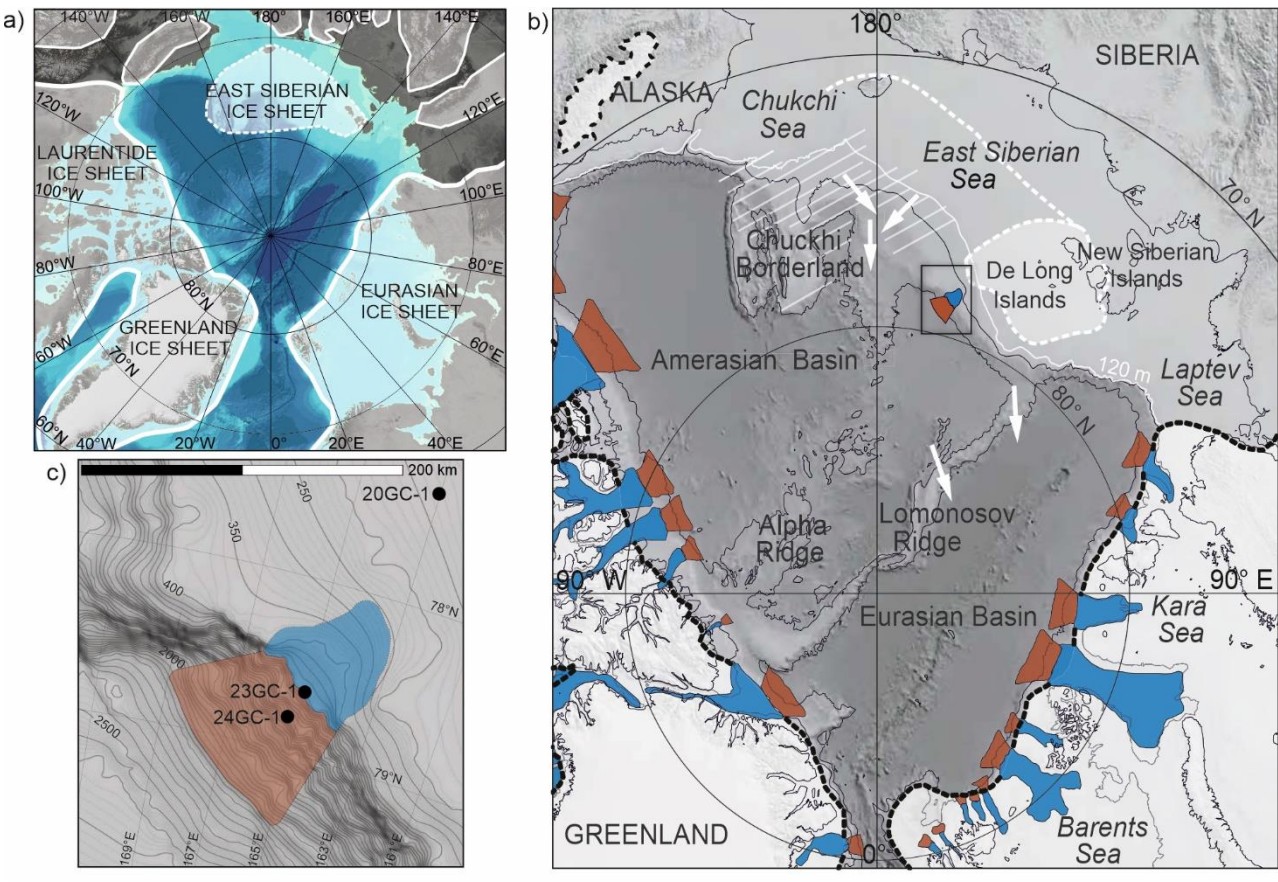

Figure 1. a) Quaternary ice sheets in the Arctic according to Jakobsson et al. (2010). Proposed maximum extent of the East Siberian Ice Sheet (MIS 6), and LGM ice in Siberia and Alaska redrawn from Niessen et al. (2013). Base map from the International Bathymetric Chart of the Arctic Ocean (IBCAO) by Jakobsson et al. (2012). b) The extent of Quaternary glaciations in the Arctic region is shown by black and white dashed lines according to Jakobsson et al. (2014). The possible extent of exposed land during the global eustatic low stand of the LGM is highlighted with the 120 m isobath across the Chukchi and East Siberian seas (O'Regan et al., 2017). White arrows indicate the direction of ice flow by Jakobsson et al. (2016) and glacial extent around the De Long Islands redrawn from Basilyan et al. (2008), respectively. Glacially excavated cross-shelf troughs (blue) and trough mouth fans (brown) are according to Batchelor and Dowdeswell (2014), and the single trough (De Long Trough) on the East Siberian shelf by O'Regan et al. (2017). Black rectangle marks the location of the map shown in c. Original figure by O'Regan et al. (2017). c) Insert figure showing the locations of studied cores 20-GC1, 23-GC1 and 24-GC1, while the brown and blue areas illustrate the constraints of the De Long glacial trough and trough mouth fan deposits (O'Regan et al., 2017). Original figure by O'Regan et al. (2017).

The occurence and direction of glacigenic features and sedimentary deposits on the lower slope of the ESS, on the seafloor in the Amerasian Basin, and on the crest of shallower plateaus and ridges offer evidence of glacial ice on the Siberian continental shelf (Niessen et al., 2013; Jakobsson et al., 2016). The direction of streamlined glacial lineations on the base of the East Siberian continental slope andArlis Plateau seabeds (Fig. 1) indicate ice flow from the East Siberian shelf (Niessen et al., 2013; Jakobsson et al., 2016). Grounded glacial ice flowing from the East Siberian shelf is also indicated by glacial lineations on an ice-scoured crest of the southern LR, along with a steep lee side towards the Amundsen Basin and a lightly sloping stoss side facing the Makarov Basin (Jakobsson et al., 2016).

### 2.3 Geological provinces and parent rocks for heavy minerals

Heavy mineral assemblages can be used to define possible parent rocks and provide insights into the overall sediment provenance and transport history in cases when reworking and mixing of several sources have occurred before deposition. The heavy minerals from the East Siberian shelf, DLT and southern LR are interpreted by considering diverse sediment transport pathways e.g., major river transport as well as parent rocks in hinterland. The heavy mineral assemblages are compared with published data from eastern Siberian geological provinces (Fig. 2) and their major rocks and heavy minerals (Table 1). According to this comparison and a closer look at the geochemical compositions of individual heavy minerals, it is possible to identify the main parent rocks for the studied DLT and southern LR sediments. The geology of the hinterland areas is characterised by the Archean and Paleoproterozoic Shield Complex called the Omolon massif, Paleozoic–Mesozoic orogenic fold belts including the Chukotka Fold Belt and the Verkhoyansk Fold Belt, Mesozoic–Cenozoic Okhotsk–Chukotka Volcanic Belt and related sedimentary rocks, and Paleozoic–Mesozoic platforms as well as the ESS sediments. The Verkhoyansk Fold Belt along with ChukotkaFold Belt are major metamorphic sources. According to Prokopiev et al. (2009) Early to Middle Paleozoic carbonate and terrigenous rocks are predominantly actinolite-chlorite, sericite-chlorite, epidote-actinolite-chlorite, and carbonate-sericite-chlorite-quartz-albite schists in their present metamorphic form. The Okhotsk–Chukotka Volvanic Belt can be considered as a major igneous and volcanic source. It includes Paleozoic–Late Cretaceous granitoids and gabbroic intrusions and Early Jurassic–Late Cretaceous basaltic andesites and rhyolite tuffs and ignimbrites (Kabanova et al. 2011). Tikhomirov et al. (2012) divided the stratigraphy of the OkhotskChukotka Volcanic Belt into three main components as follows; the lower andesites, the group of formations dominated by silicic rocks and the upper basalts.

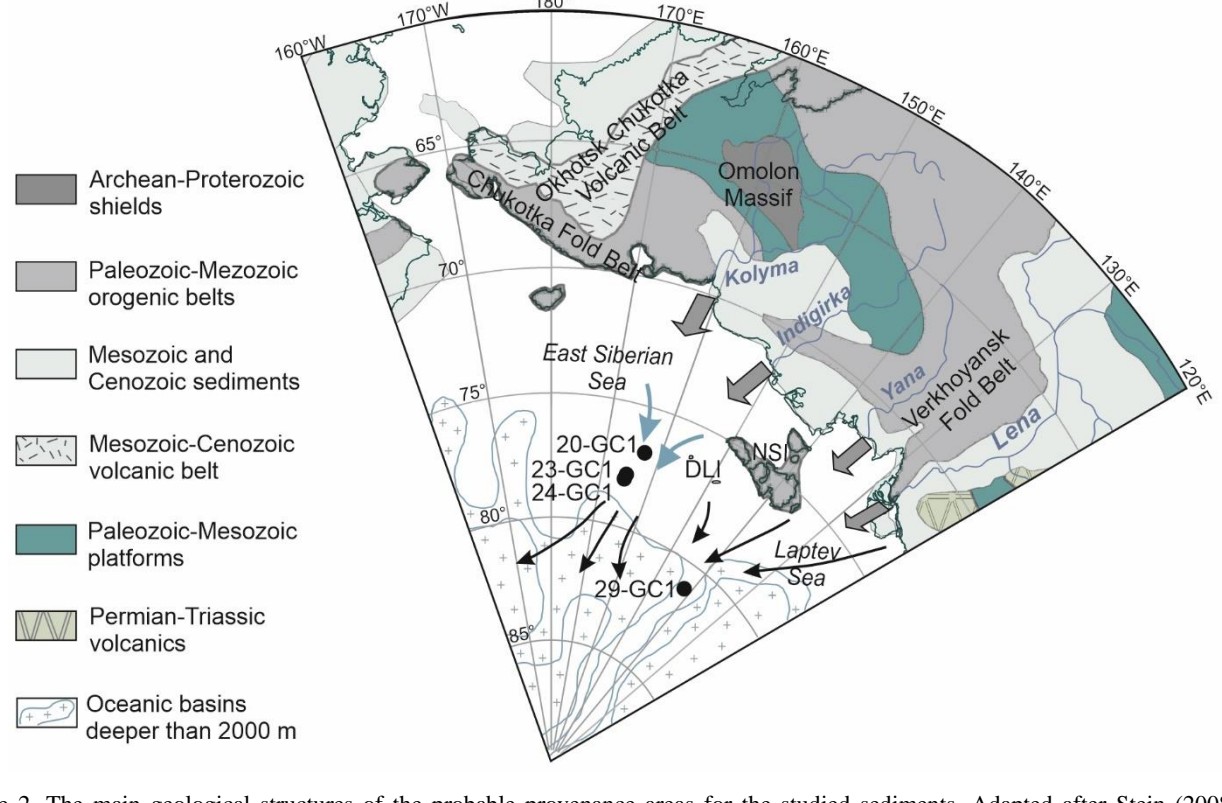

Figure 2. The main geological structures of the probable provenance areas for the studied sediments. Adapted after Stein (2008) and Kaparulina et al. (2016). Grey arrows indicate sediment transport from the rivers, blue arrows present inferred ice flow, and black arrows indicate inferred ice rafting. DLI – De Long Island, NSI – New Siberian Islands.

Table 1. A summary of lithological and mineralogical characteristics of the source areas and rocks for the De Long Trough and southern Lomonosov Ridge sediments.

| Geological province or area | Geology and lithologies of source rocks | Heavy minerals |
|---|---|---|
| *Omolon Massif [1,2,3]* | Archaean and Proterozoic magmatic and metamorphic rocks; granulites (high grade metamorphic rocks), gneisses, quartzites, and granites | Major: garnet, amphiboles (hornblende), hypersthene<br>Minor: magnetite, sapphirine-spinel |
| *Verkhoyansk Fold Belt4 [4,5,6]* | Paleozoic shales and carbonate rocks, Mesozoic sand-siltstones and black shales (mid- to low-grade metamorphic rocks) | Major: amphiboles (hornblende, actinolite), garnet, Fe-oxides<br>Minor: dolomite, epidote |
| *Chukotka Fold Belt [7,8]* | Paleozoic shales and carbonate rocks, Mesozoic sand-siltstones, and shales, some granitoids (mid- to low-grade metamorphic rocks); Triassic gabbroic rocks, some Cretaceous volcanogenic rocks | Major: amphiboles (hornblende, actinolite), biotite, apatite,<br>Minor: sillimanite, andalusite, sphene, zircon |
| *Siberian platform [9]* | Paleozoic–Mesozoic volcanic–sedimentary rocks; Jurassic sand-siltstones and tuffaceous rocks | Major: Fe-oxides, pyroxenes, hornblende<br>Minor: zircon |
| *Okhotsk-Chukotka Volcanic Belt [10, 11, 12, 13, 14]* | Paleozoic-Early Mesozoic sandstones, Paleozoic-Late Cretaceous granitoid and gabbroic rocks, Early Jurassic-Late Cretaceous calc-alkaline volcanic rocks, and felsic ignimbrites | Major: pyroxenes, amphiboles, garnet, epidote, biotite<br>Minor: magnetite, ilmenite, apatite, sphene, zircon, olivine, monazite |
| *East Siberian Sea [15, 16]* | Silty-clay, clayey-silt, sandy-silty-clay associated with ice rafting. Sources of the sediments of the East Siberian Sea are the Indigirka and Kolyma rivers and the New Siberian Island region. The inner shelf is characterised by low concentrations of heavy minerals. | Major: amphiboles (hornblende, actinolite) pyroxenes, Fe-oxides, ilmenite, biotite, epidote<br>Minor: zircon, sphene, anatase, rutile |

*References:* 1) Akinin and Zhulanova (2016), 2) Avchenko and Chudnenko (2020), 3) Drachev (2016), 4) Kaparulina et al. (2016), 5) Konstaninovsky (2007), 6) Prokopiev et al. (2009), 7) Miller et al. (2009), 8) Katkov et al. (2007), 9) Filatova and Khain (2008), 10) Akinin and Miller (2011), 11) Belyi (1977), 12) Kabanova et al. (2011), 13) Tikhomirov et al. (2012), 14) Tschegg et al. (2011), 15) Naugler et al (1974), 16) Nikolaeva et al. (2013).

10    **3 Materials and methods**

The samples studied in this paper are from four sediment cores collected during Leg 2 of the SWERUS-C3 2014 expedition on the Ice Breaker *Oden,* which departed on 21 August from Barrow, Alaska and ended on 3 October in Tromsø, Norway (Fig.

3). The samples are from cores 20-GC1, 23-GC1, 24-GC1 and 29-GC1, which were collected from various locations and water depths (Table 2).

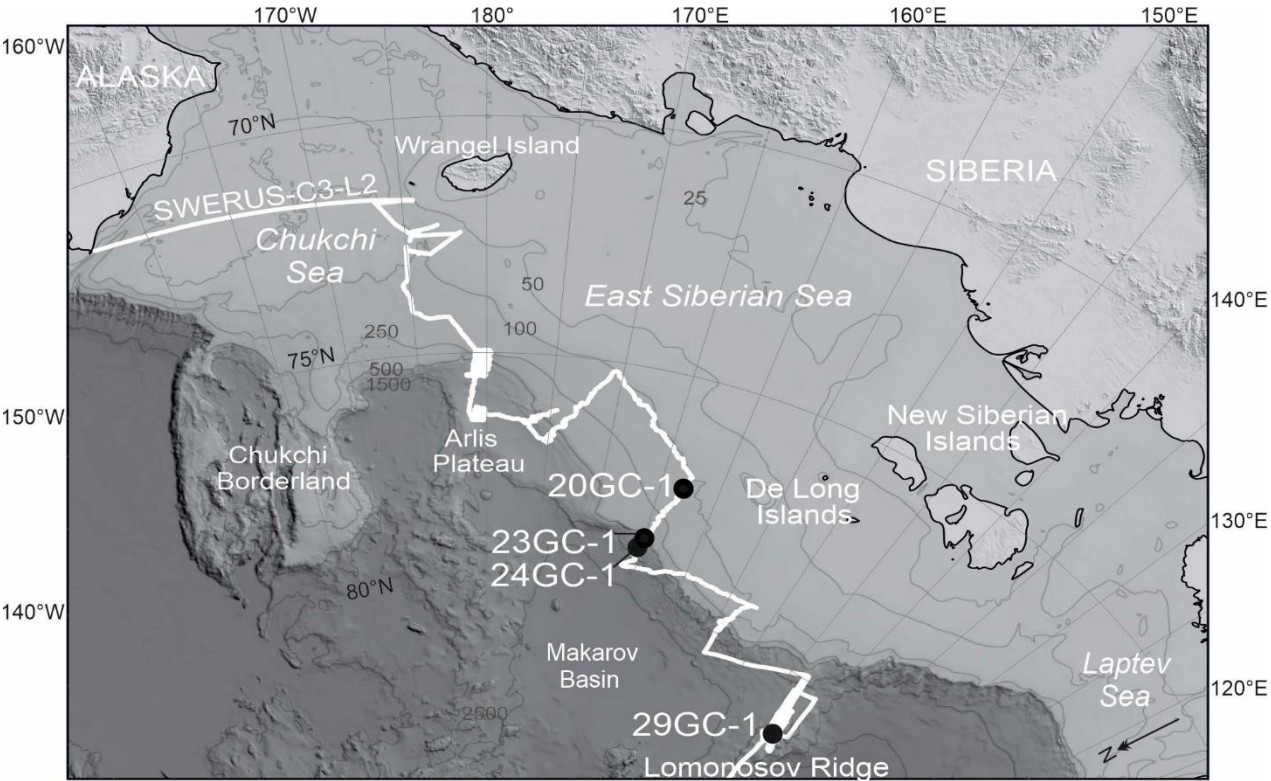

Figure 3. A map showing the locations of studied cores 20-GC1, 23-GC1, 24-GC-1 and 29-GC1, and the route of Leg 2 of the SWERUS-C3 2014 expedition (white line). Original map by O'Regan et al. (2017).

Table 2. Locations, lengths, and water depths of the cores studied. Data from O'Regan et al. (2017) and the SWERUS cruise report (2014).

| Core | Water depth | Length (m) | Latitude | Longitude | Location |
|------|-------------|------------|----------|-----------|----------|
| 20–GC1 | 115 | 0.83 | 77°21.5' N | 163°2.0' E | East Siberian Shelf |
| 23–GC1 | 508 | 4.06 | 78°39.7' N | 165°0.9' E | De Long Trough |
| 24–GC1 | 964 | 4.05 | 78°47.8' N | 165°22.0' E | De Long Trough |
| 29–GC1 | 824 | 4.66 | 81°17.9' N | 141°46.9' E | Southern Lomonosov Ridge |

## 3.1 Sedimentary and acoustic units

The cores studied were collected using a gravity corer (GC), and the geophysical mapping methods during the expedition included multibeam bathymetry and sub-bottom profiles (Jakobsson et al., 2016; O'Regan et al., 2017). Physical property measurements were carried out with a Multi-Sensor Core Logger (MSCL)and the cores were split, described, and imaged shipboard. and the grain size, x-ray fluorescence (XRF) core scanning, and additional magnetic susceptibility were done at the Department of Geological Sciences, Stockholm University (O'Regan et al., 2017).

According to O'Regan et al. (2017) the sub-bottom stratigraphy from the shallow East Siberian shelf to the shelf break is divided into six acoustic units (AU), and two of the studied cores (23-GC1 and 24-GC1) are divided into two sedimentary units A and B (Fig. 4). The division and correlation between sedimentary units is based on grain size and physical properties data and incorporation of the Ca/Ti ratio from the XRF-scanning data. AU 1 is interpreted as iceberg-scoured postglacial sediments that has a sharp basal contact on the shallow shelf. In deeper water depths, the unit thickens and incorporates preglacial and glacial sediments reworked by the last glacial cycle sea level lowering. Core 20-GC1 sampled sediments from

this unit. AU 2 is interpreted as outcropping sedimentary or bedrock strata with no knowledge of its age or composition. AU 3 is a coherent and laterally continuous acoustically layered sequence extending seaward of the shelf break and downslope to water depths of > 2000 meters below sea level (m b.s.l.). Cores 23-GC1 and 24-GC1 penetrated to the base of AU 3, which sediments are represented by sedimentary Unit A in the cores. These undisturbed sediments overlie the coarser-grained glacial sediments of sedimentary Unit B, corresponding to AU 4 which is, along with AU 5 interpreted as sub-glacially deposited sediments. The bases of these units are separated at the shelf break by AU 6, which is interpreted as either a mass wasting deposit or an ice-proximal fan.

## 3.2 Sediment cores and heavy mineral samples

A total of 17 samples were analysed in this study (Table 3), and their locations in relation to the sediment stratigraphy are illustrated using the digital images of the cores (Fig. 4). Two samples are from the 0.83 m long core 20-GC1, which was recovered in 115 m water depth from the East Siberian shelf, near the DLT. The core contains dark grey coarser-grained facies at its base and sediments with lower-density and susceptibility above 0.5 meters below sea floor (m b.s.f.). Two diamict samples are from the 4.06 m long core 23-GC1, which was recovered in water depth of 508 m, and nine samples are from 4.05 m long core 24-GC1 recovered in water depth of 964 m from the DLT. The samples from core 24-GC1 also include surface sediments and other lithologies found between the seafloor and the diamict. These samples help define the mineralogy of the diamict in comparison to the overlying sediments from the last glacial cycle. Four samples are from core 29-GC1. The 4.66 m long core was recovered in 824 m water depth from the southern LR, and the core sampled acoustically stratified sediments deposited on top of the ice-scoured surface (Jakobsson et al., 2016). The base of this core – represented here by three samples – recovered a dark grey diamict whose mineral assemblage is compared to the DLT diamict samples

Table 3. Depth, cm interval and description of the 17 studied samples from cores 20-GC1, 23-GC1, 24-GC1 and 29-GC1.

| Sample | Core | Section | Depth | cm | Description | Location |
|---|---|---|---|---|---|---|
| 1 | 20–GC1 | CC | 0.76 | 22-24 | grey diamict | East Siberian shelf |
| 2 | 20–GC1 | CC | 0.81 | 27-29 | grey diamict | East Siberian shelf |
| 3 | 23–GC1 | 3 | 3.66 | 108-110 | grey diamict | De Long Trough |
| 4 | 23–GC1 | 3 | 3.86 | 128-130 | grey diamict | De Long Trough |
| 5 | 24–GC1 | 1 | 0.04 | 3-5 | dark brown clay | De Long Trough |
| 6 | 24–GC1 | 1 | 0.53 | 52-54 | light brown silty clay | De Long Trough |
| 7 | 24–GC1 | 1 | 0.99 | 98-100 | olivegreen-grey silty clay | De Long Trough |
| 8 | 24–GC1 | 2 | 1.53 | 47-49 | olivegreen-grey silty clay | De Long Trough |
| 9 | 24–GC1 | 2 | 2.09 | 104-106 | light brown silty clay | De Long Trough |
| 10 | 24–GC1 | 3 | 2.88 | 34-36 | grey clay above diamict | De Long Trough |
| 11 | 24–GC1 | 3 | 3.12 | 58-60 | grey clay above diamict | De Long Trough |
| 12 | 24–GC1 | 3 | 3.61 | 108-110 | grey diamict | De Long Trough |
| 13 | 24–GC1 | 3 | 3.77 | 122-124 | grey diamict | De Long Trough |
| 14 | 29–GC1 | 4 | 4.31 | 114-116 | light brown silty clay | Southern Lomonosov Ridge |
| 15 | 29–GC1 | 4 | 4.43 | 126-128 | grey diamict | Southern Lomonosov Ridge |
| 16 | 29–GC1 | 4 | 4.55 | 138-140 | grey diamict | Southern Lomonosov Ridge |
| 17 | 29–GC1 | 4 | 4.62 | 145-147 | grey diamict | Southern Lomonosov Ridge |

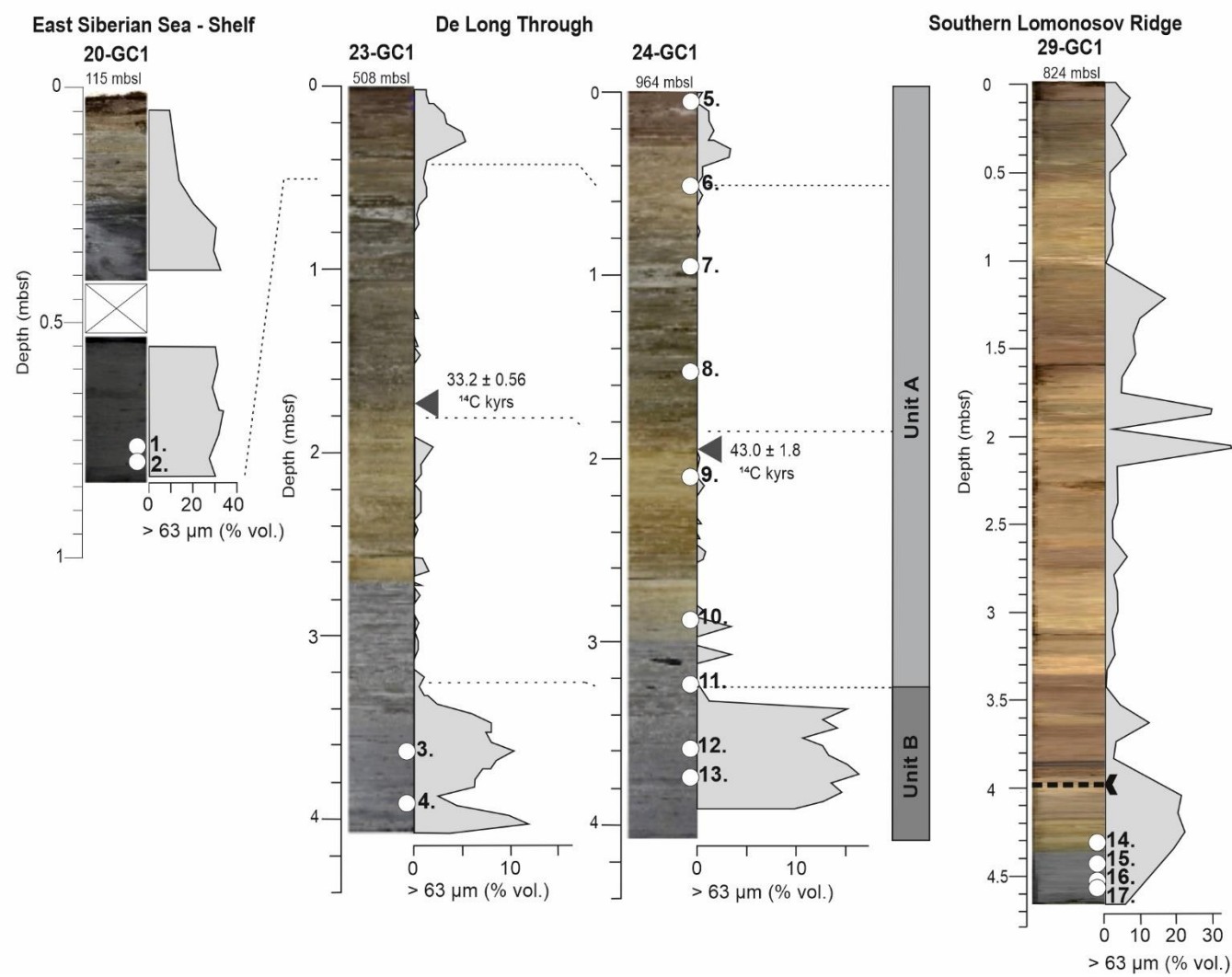

Figure 4. Digital images and locations of studied cores, including 17 sample points (white circles) (adapted from O'Regan et al., 2017). Stratigraphic correlation of the cores (dotted lines) and the division of sedimentary Units A and B are based on MSCL, XRF-scanning, grain size, digital images and radiocarbon dating results (O'Regan et al. 2017). Inferred MIS 6/5 transition (black dotted line) in core 29-GC1 from Jakobsson et al. (2016) and grain size from West et al. (2021).

### 3.3 Dating and chronology

Six radiocarbon ages were obtained from the core catcher of 20-GC1, and single range finding ages were obtained from both 23-GC1 and 24-GC1 (Cronin et al., 2017; O'Regan et al., 2017). The accelerator mass spectrometry (AMS) radiocarbon measurements were performed at the Lund University Radiocarbon Dating Laboratory (Lu), Sweden, and at the National Ocean Sciences Accelerator Mass Spectrometry (NOSAMS) facility at Woods Hole Oceanographic Institution, Massachusetts on samples containing the planktonic foraminifer *Neogloboquadrina pachyderma*, mixed benthic foraminifera or mollusk shells. Radiocarbon dates from the lower part of core 20-GC1 outlined ages between ~ 13 and 11 ka, and also indicate that the dense, deglacial, dark grey sediments in the lower part of 20-GC1 do not correlate to a similar lithology, predating the LGM, at the base of cores 23-GC1 and 24-GC1 (Cronin et al., 2017). The sediments in the lower part of core 20-GC1 correlate with a thin, dense, higher-susceptibility grey sediment sequence in the upper 0.5 m of cores 23-GC1 and 24-GC1 (Fig. 4). Radiocarbon dating from cores 23-GC1 (1.69–1.86 m b.s.f.) and 24-GC1 (1.92 m b.s.f.) give ages of 33 200 ± 560 and 43 000 ± 1800 $^{14}$C years BP, and the calibrated medium ages are 37 000 $\pm\frac{2600}{1300}$ and 46 300 $\frac{3500}{2600}$ cal years BP (O'Regan et al., 2017). Based on the results from radiocarbon dating, the base of Unit A is older than ~ 50 cal kyr BP. Occurrence of the calcareous nannofossil *E. huxleyi* in core 23-GC1 at 2.28 m b.s.f. indicates, that the sediments in that point are younger than MIS 7/8

(Backman et al., 2009; O'Regan et al., 2017; O'Regan et al., 2020). O'Regan et al (2017) argued for an MIS 6 age for the basal diamict in 23-GC1 and 24-GC1, but acknowledged that it may have been deposited during a younger stadial in MIS 5 or 4. According to Jakobsson et al. (2016), Holocene age is indicated by calcareous nannofossils in the uppermost 5 cm of core 29-GC1, and an observation of *E. huxleyi* was made at 3.81 m b.s.f. The core correlates to a neighbouring core which base has been proposed to be MIS 6 age by Stein et al., 2001. The inferred MIS 6/5 transition in core 29-GC1 is at ~ 4 m b.s.f. (Jakobsson et al., 2016). ). The results of luminescence dating by West et al. (2021) support a pre-Eemian age for the base of this core.

**3.4 Heavy mineral analysis**

To clean the samples from adhering clays, the samples were weighed and treated with 2 ml of dispersant solution, sodium pyrophosphate ($Na_4P_2O_7$) and distilled water. The samples were put in an ultrasonic cleaner for 30–45 minutes and occasionally stirred with a glass rod. To separate the silt from the coarser fraction, the samples were wet-sieved into a fraction coarser than 63 μm and then dried and collected for heavy liquid separation. Nine samples had enough material for the heavy liquid separation with sodium heteropolytungstate (LST Fastfloat) with a density of 2.82 g/cm$^3$. The >63μm fraction was poured into a separating funnel consisting of heavy liquid. The mixture was shaken thoroughly and left to allow the heavy minerals to separate from the light minerals. After separation, the heavy fraction was drained onto a filter paper and cleansed with distilled water using a Büchner funnel for suction filtration. The filter paper with the heavy fraction was dried at 60 °C and the dry fraction collected. Heavy mineral separation was done for six samples with liquid nitrogen, following the method of Mange and Maurer (1992). Heavy mineral separation was not done for two samples due to the low amount of material. The heavy mineral samples for the electron microscope were prepared in the thin section laboratory at Oulu Mining School, University of Oulu.

The heavy mineral grains >63 μm from 17 samples were analysed with Zeiss Ultra Plus Scanning Electron Microscope (SEM) at the Centre for Material Analysis, University of Oulu. Heavy mineral identification was done by using MinIdent-Win 3.0 computer program. The 18 analysed elements Na, Mg, Al, Si, P, S, K, Ca, Ti, V, Cr, Fe, Cu, Zn, Zr, La, Ce, and Nd (expressed as oxides) were used in the specification of the minerals.

**4 Results**

**4.1 Sediment stratigraphy**

The lithostratigraphic definition and stratigraphic correlation of the cores 20-GC1, 23-GC1 and 24-GC1 are based on digital images, MSCL, XRF-scanning, grain size and radiocarbon dating results by O'Regan et al. (2017) (Fig. 4). Core 20-GC1 was obtained from a shallower shelf containing dark grey finer-grained deglacial sediments with coarser-grained facies at its base. The bases of 23-GC1 and 24-GC1 (sedimentary unit B) also contain a dark grey and poorly sorted sequence of coarser-grained diamict, which is interpreted to be formed by grounded glacial ice flowing out from the East Siberian shelf, and there is a less pronounced coarser-grained interval present in the upper 50 cm of these cores. The surface sediments and lithologies from the last glacial cycle, which were found between the seafloor and the diamict in core 24-GC1, include dark and light brown clay, olive green-grey silty clay, and grey clay. Core 29-GC1 is dominated by light brown to dark brown sediments and the base of the core also has dark grey diamict sequence overlain by light brown silty clay. The diamict sequence could be associated with the extensive scouring of the LR but could also be related to the initiation of deglaciation.

## 4.2 Heavy minerals

The heavy minerals presented in this study are the major silicates including amphiboles, pyroxenes, epidote, garnets, micas, titanite and zircon, and other heavy minerals including Fe-oxides, ilmenite, other oxides (anatase, crichtonite, armalcolite), phosphates, sulphides (kieserite, voltaite) and others (vesuvianite, olivine, neptunite, chlorite). The abundance of each identified mineral is presented as a relative percentage of the total grains counted in each sample. The total amount of identified grains varies throughout the samples and the quantity of total grains (n) is low in a few samples due to the scarce amount of grains >63μm and the lack of heavy minerals.

The comparison of heavy mineral assemblages between different diamict samples and deglacial sediment samples can be considered after mineral identification based on geochemical compositions (Fig. 5). It is especially important to compare the southern LR diamict assemblages in core 29-GC1 (samples 15-17) with the diamict assemblages in the DLT samples from cores 23-GC1 and 24-GC1. Two interglacial samples from core 20-GC1 (1-2) are close to DLT, and one light brown silty clay sample (14) overlays dark grey diamicts at the base of core 29-GC1 from the southern LR (Fig. 5). The heavy mineral assemblages of the samples from core 20-GC1 consist of major amphiboles (22–43%), pyroxenes (15%), epidote (7–11%), and minor garnets (3–14%), and titanite (~10%), with a distinct occurrence of zircon, Fe-oxides and ilmenite, along with other oxides, phosphates, and others with a concentration between 1 to 8%. In core 23-GC1, the heavy mineral assemblages consist mainly of amphiboles (24%), pyroxenes (~17%), and epidote (11–16%), and other minerals with a concentration varying from 1 to 11%. In core 24-GC1, amphiboles (33–39%), pyroxenes (17–21%), and epidote (7–17%) are also main contributors to the heavy mineral assemblages, whereas minor minerals are present in smaller amounts varying from 1 to 6%. Core 29-GC1 has the highest amphibole content varying between 33 and 50%. Pyroxenes (7–14%) and epidote (8–20%) content is relatively lower, but these are also the main contributors. The presence of minor minerals varies between 1 and 10%. There is a prominent increase in amphibole content in core 29-GC1 and a lower pyroxene content compared to the other cores. In comparison to cores 23-GC1 and 24-GC1, there is also a slight increase in garnet content in core 29-GC1. Core 20-GC1 has the highest titanite, zircon, and ilmenite content.

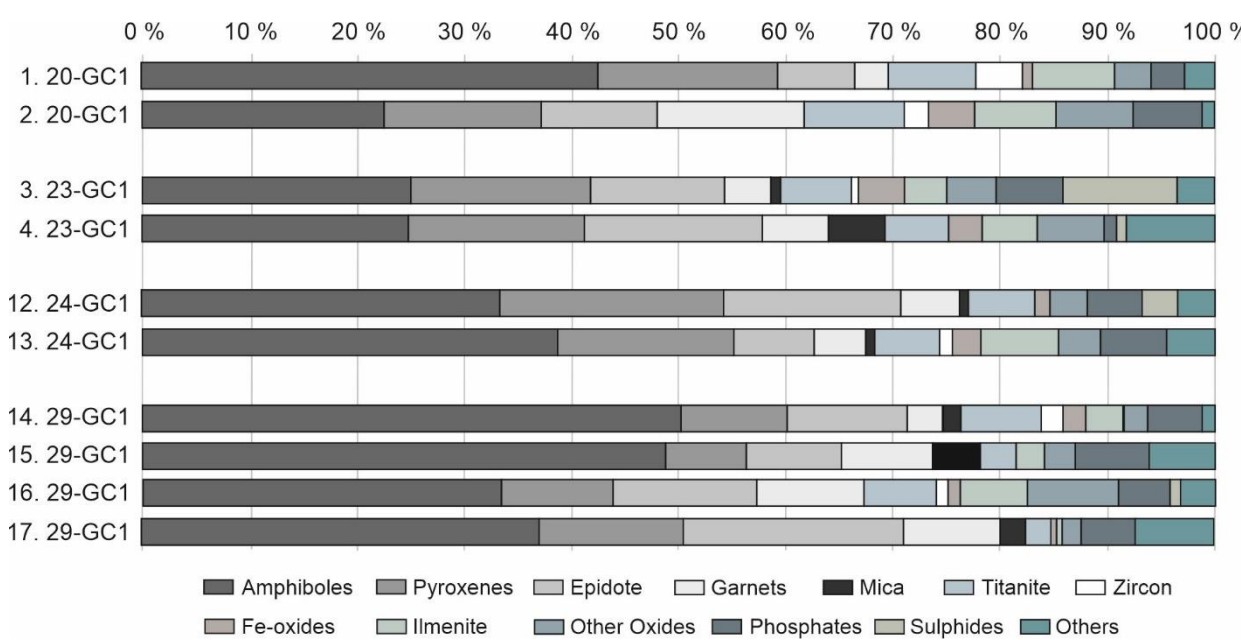

Figure 5. The distribution of heavy minerals of diamict and deglacial sediments of the samples studied. Samples 1 and 2 are deglacial sediments from the East Siberian shelf, and their depositions correlate to the sediments from the last glacial cycle in the upper part of cores 23-GC1 and 24-GC1. Samples 3, 4, 12, and 13 are diamict from the base of cores 23-GC1 and 24-GC1 from the De Long Trough. Sample 14 is deglacial clay, and samples 15–17 are diamict from the base of core 29-GC1 from the southern Lomonosov Ridge.

From core 24-GC1, five samples (5, 6, 9, 10, and 11) of surface sediments and other lithologies were selected to compare their heavy mineral assemblages with the underlying diamicts (12 and 13) and to detect possible stratigraphic variability in heavy mineral assemblages along the core (Fig. 6). Two of the treated samples (7 and 8) did not have enough coarse fraction grains or heavy minerals to be analysed. It should also be noted that the heavy mineral grain content in samples 5 and 11 is notably lower (n = 24-36) than in the other samples. The diamicts from samples 12 and 13 consist mostly of amphiboles (33–39%), pyroxenes (17–21%), and epidote (7–17%), with minor mineral composition ranging between 1 and 7%. In comparison to these diamicts, there are less amphiboles (8–28%), a varying number of pyroxenes (6–23%), and epidote (8–20%) in the interglacial sediment samples. There is a notable increase of garnet content in sample 11, and it is the most abundant mineral in the sample. Although minor minerals are present in the assemblages varying from 1 to 11%, peaks appear in the concentrations of other heavy minerals (incl. chlorite) in sample 5 (19%) and of phosphates in sample 10 (14%). Overall, the amphibole content is higher in the diamicts (up to 39%) than in the interglacial sediments (up to 28%). The pyroxene concentration is smaller and fluctuates more in the interglacial sediments than in the diamicts. Epidote concentration is similar in the diamicts and overlying sediments. Minor minerals are slightly more abundant in the interglacial sediments than in the diamicts, in addition to the previously mentioned peaks in samples 5 and 10. The overlying interglacial sediments show an increase in iron oxide content right above the diamict and in the uppermost part of the core. There is also an increase in other oxides in the middle part of the core, and in the titanite content upwards from the middle part of the core.

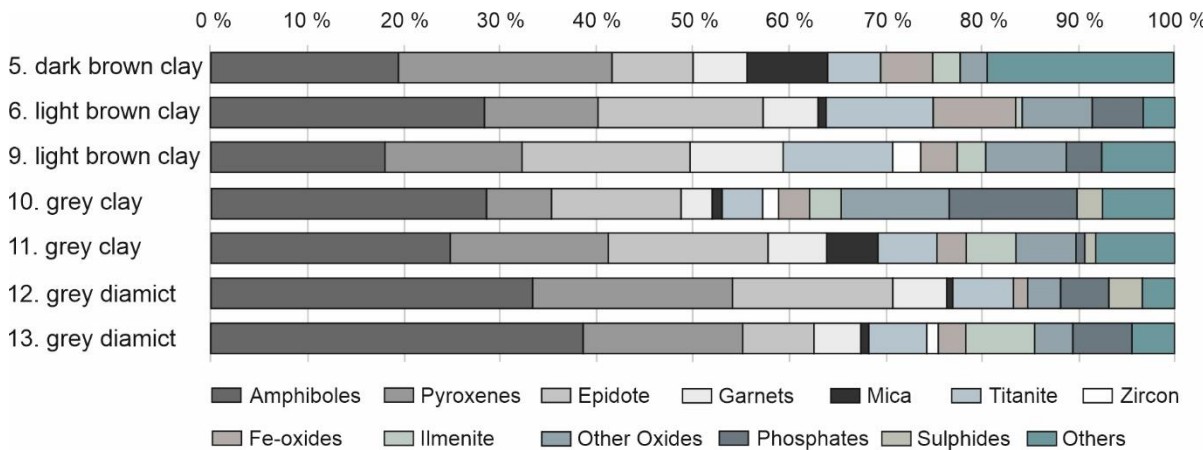

Figure 6. The distribution of heavy minerals of sediments from core 24-GC1 from the De Long Trough. Samples 12 and 13 are grey diamict from the base of the core. Samples 9, 10 and 11 are interglacial sediments from the middle part of the core, and samples 5 and 6 are from the upper part of the core above the radiocarbon age point of 43 000 ±1800 $^{14}$C years BP.

## 5 Discussion

This provenance study interprets the parent rocks and sediment source areas of glacigenic deposits related to the ESIS distribution on the East Siberian continental margin Parent rocks determine the basics for heavy mineral assemblages, and since sedimentary processes are governed by glacial ice dynamics, it can be assumed that changes in mineralogy reflect also changes in sediment sources and ice streaming before final deposition and burial. The distribution of heavy minerals in the sediments is influenced by the differention between sub-ice and subglacial shelf, and open-marine sedimentary processes. During glacial and deglacial intervals iceberg-rafted sediments with a high content of coarse-grained material are common (Stein et al., 2012; O'Regan et al., 2014), and in contrast, the suspension of finer-grained material consolidates in sea ice, especially on the Siberian shelves (Darby et al., 2009). Sea ice rafting is most likely the primary sediment transportation mechanism (Darby et al., 2009; Polyak et al., 2010) during interglacial/interstadials, and sediments with an already mixed

composition can be delivered mostly from the continental margins instead of directly from the inner terrestrial regions (Dong et al., 2020).

As concluded by Behrends et al. (1999), the important factors controlling sediment supply along the East Siberian continental margin includes fluvial input, sea ice, and ocean currents, and the sediment input onto the shelf is controlled by the major Siberian River systems. According to Naugler et al. (1974), the primary riverine sources for the ESS sediments are the Indigirka and Kolyma rivers. The heavy mineral assemblages of the sediments occurring in the paleovalley of the Indigirka River have higher abundances of amphiboles with relatively high proportions of titanite and ilmenite, in comparison to the Kolyma River paleovalley sediments which show a higher share of pyroxenes and epidote (cf. Nikolaeva et al., 2013). The shallow shelf of the LS is considered a notable source area of terrigenous material with higher content of amphiboles and pyroxenes, and the amphibole in the eastern part of the LS is supplied by the Yana and Lena rivers from the Siberian hinterland (e.g., Behrends et al., 1999; Schoster et al., 2000). Viscosi-Shirley et al. (2003) have stated that, for example, the shale-rich ESS and eastern LS sediments are mostly derived from the Omolon massif and Verkhoyansk Fold Belt mountains that lie within the drainage basins of these rivers, and the Kolyma River's drainage area intersects also with the Okhotsk–Chukotka Volcanic Belt.

The present study indicates that the Indigirka and Kolyma River sediments were a major contributor to the materials eroded and re-deposited by glacial ice in the DLT. The mineral content variability of these rivers' sediments can be seen within the heavy mineral assemblages of the DLT diamict. A distinct feature in these assemblages is the elevated content of minerals characteristic of ancient volcanogenic and volcanogenic-sedimentary rocks of greenstone metamorphism enriched in amphibole, pyroxenes, and epidote (Fig. 5). The mineralogical signature of the DLT diamicts consisting predominantly of amphiboles and pyroxenes with minor content of garnet and epidote show a clear delivery from the eastern part of the ESS. The physical properties of the DLT glacial diamict closely resemble a pervasive diamict unit recovered across the southern LR, but according to their heavy mineral content, their source material is slightly different. The assemblages from core 29-GC1 from the southern LR have elevated amphibole and garnet content, along with moderately higher titanite and ilmenite content, which emphasise the Verkhoyansk Fold Belt as a possible source. This possibly indicates material delivery by ice rafting and large-scale iceberg inputs from westerly sources, such as the LS. The high amphibole content (>30%) of the DLT and southern LR sediments is characteristic for the ESS, and eastern LS sediments documented by e.g., Behrends (1999) and Schoster et al. (2000).

The dark grey coarser-grained sub-glacially deposited diamict from the DLT are overlain by a unit of sediments from the last glacial cycle (Fig. 4). A shift from diamict-dominated sediments to clay-dominated sediments illustrates well the transition from glacial to interglacial conditions. It can be assumed that changes in mineralogy between the diamicts and the overlying interglacial/interstadial sediments reflect those changes related to the dynamics of the deglaciation in the ESS. The finer-grained, grey sediments just above the diamicts are most likely related to glacial maxima and followed by deglacial deposition.This is recorded by decreasing amphibole and increasing diversity in all heavy mineral contents (Fig. 6). The layer of olive green/grey fine grained sediments are characterised by the lack of grain size > 63 µm and no heavy minerals were retrieved from the studied samples of this part. The sediments were possibly derived from the inner shelf of the ESS, which is characterised by low concentrations of heavy minerals (cf. Naugler et al., 1974). These factors indicate that during the deposition of these sediments the shelf or at least the shoreline and river discharge region was possibly free from ice, or the ice sheet was relatively thin and not grounding on the shelf. The uppermost sediments also show a slight increase in Fe-oxides content which could indicate sea-ice rather than iceberg transport.

Stratigraphy and dating of sediments from the DLT show that sediments covering the glacial deposits are older than ~ 50 cal kyr BP, which offers evidence for diamict deposition and large-scale glacial activity occurring during the MIS 6, stadial in MIS 5, or the glacial period of MIS 4 (c.f., O'Regan et al., 2017). According to Ye et al. (2020), during MIS 4, the East Siberian continental margin environments were characterised by poor circulation, lack of IRD, and large volumes of fine-grained sediment input from the East Siberia shelf. Based on the geophysical mapping, the sediments recorvered from the East Siberian shelf (20-GC1) are iceberg-scoured postglacial sediments and possibly incorporated with reworked preglacial and glacial sediments. These sediments interpreted as MIS 1 age (cf., O'Regan et al., 2017) in the lower part of the core correlate with a grey sediment sequence in the uppermost part of the cores from the DLT (Fig. 4). The lithology of the sediments deposited on top of the ice-scoured surface of the southern LR (29-GC1) correlates to many of the sediment cores previously recovered around the area during various Polarstern expeditions (e.g. Rachor, 1997; Stein, 2015, 2019), and to mineralogical and geochemical data produced from the cores (Behrends, 1999; Müller and Stein, 2000; Schoster et al., 2000). The sediment cores (e.g., PS2757-8, PS87/086-3, 29-GC1 and 29-PC1) retrieved from the southern LR show excellent lithostratigraphic correlations (Jakobsson et al., 2016; Stein et al., 2017). Based on a tentative age model, the lower parts of the cores are inferred to be of MIS 6 age (cf. Stein et al., 2017), which is further supported by an age model by O'Regan et al. (2020). Age-depth model by West et al. (2021) also shows late MIS 6 age for the southern LR diamict, and that it might not be the actual glacial diamict associated with the scouring of the LR, but possibly later occured ice sheet rafting and large-scale iceberg inputs. This could refer to more SW sources of material transported dominantly by sea ice and icebergs associated to the break-up of the ice sheet during MIS 6. A continual record of the high amphibole content in the LS sediments is documented during MIS 6, and a resembling pattern is also detected during MIS 5 and 4 (Behrends, 1999). Although this supports the tentative MIS 6 age for the southern LR diamict, a later deposition is also possible.

According to Dong et al. (2017), studies of the sediment cores adjacent to the ESS margin are important also for comprehending the history of ESIS (e.g., Polyak et al., 2004, 2009; Stein et al., 2012). Dong et al., (2017) show that inputs from the ESIS can also be inferred from peaks of smectite, kaolinite, and chlorite associated with coarse sediment. Based on clay mineral and IRD distributions, Ye et al. (2020) suggests a further grounded ice shelf originating from the East Siberian margin over the southern Mendeleev Ridge, and the ice shelf seems to have been a part of the larger mass of the ESIS. O'Regan et al. (2017) have concluded that the presence of glacial sediments and glacio-tectonic features on the New Siberian Islands and lower slope of the ESS, along with the glacially scoured DLT on the outer margin of the shelf offer obvious evidence for glacial ice on the East Siberian continental shelf and also a wider extent of the ESIS. The orientation of glacio-tectonic features illustrates that glacial ice likely nucleated over the De Long Islands and flowed on the New Siberian Islands from a north-northeast direction (Fig. 1b; e.g., Basilyan et al., 2008; O'Regan et al., 2017). In addition, the orientation of streamlined glacial lineations on the base of the East Siberian continental slope and on heavily ice-scoured crest of the southern LR are an indication of grounded glacial ice flowing from the East Siberian shelf (Niessen et al., 2013; Jakobsson et al., 2016). These features, along with the content of the presently studied heavy mineral assemblages suggests that glacial ice not only grew out from the East Siberian shelf but also from the De Long Islands, and that there were also ice rafting and sediment transportation from westerly sources, such as the LS. (Fig. 7.).

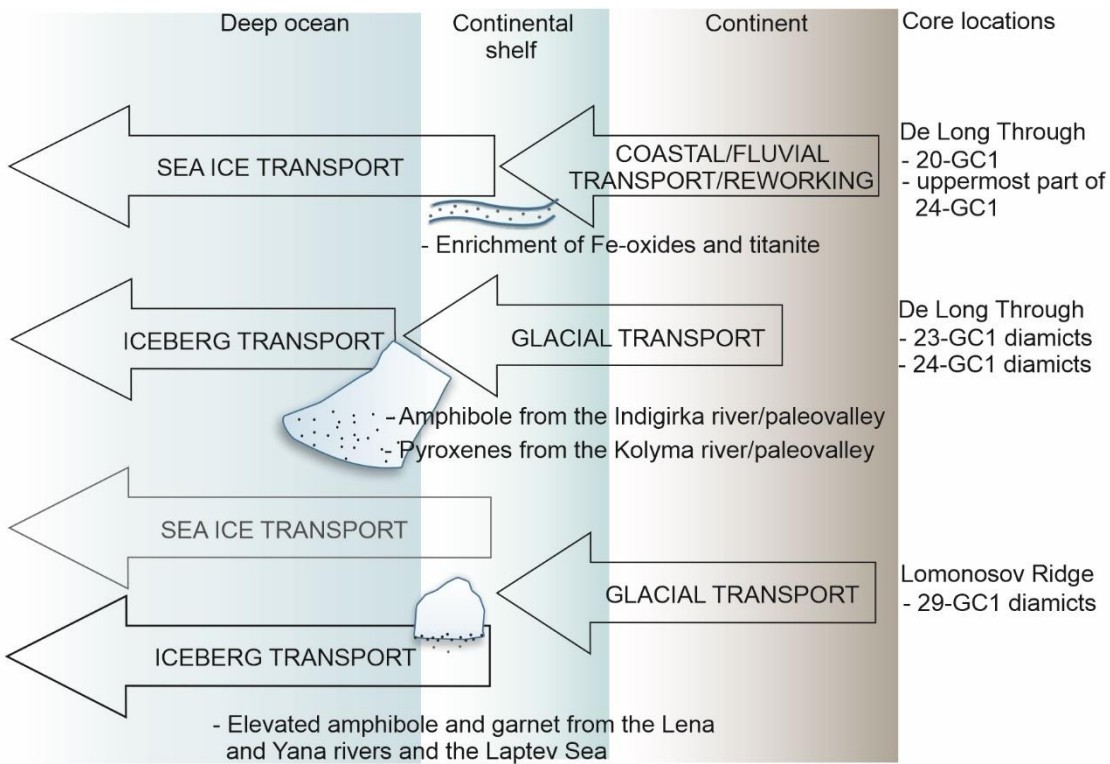

Figure 7. Synthesis of the heavy mineral analysis and interpretations of this study related to transportation and depositional sites.

The locations of the cores studied here are in the neighborhood of the potential sites of the IODP Expedition 377 (Arctic Ocean Paleoceanography – ArcOP) scheduled for autumn 2022, which key objectives are reconstruction of provenance, source areas and transport mechanisms of terrigenous sediment fractions as well as the history of the Pliocene-Pleistocene Eurasian and East Siberian ice sheets (Stein et al., 2021). In addition to the previous studies e.g., Polarstern expeditions (Rachor, 1997; Stein, 2015, 2019), the results of this study can provide also useful knowledge in the interpretation of the future IODP data.

**6 Conclusions**

The sediments from glacially scoured cross-shelf trough, the De Long Trough, extending to the edge of the continental shelf offer additional evidence that glacial ice existed on parts of the East Siberian Sea during previous glacial periods MIS 6 and 4. The data from the heavy mineral analysis and interpretations presented here provide new insights that sediments delivered to the East Siberian shelf by the Indigirka and Kolyma river are a primary source for sediments eroded and re-deposited by glacial ice in the De Long Trough. The mineralogical signature of the diamicts from the De Long Trough consists predominantly of amphiboles and pyroxenes with minor content of garnet and epidote and illustrate a clear delivery from the eastern part of the East Siberian Sea. The physical properties of the De Long Trough glacial diamict closely resemble a pervasive diamict unit recovered from the southern Lomonosov Ridge, but the heavy mineral content indicates slightly different source material. Assemblages with elevated amphibole and garnet content, along with moderately higher titanite and ilmenite content of the diamict from the southern Lomonosov Ridge, accentuate the Verkhoyansk Fold Belt as a possible source area. The results from this heavy mineral analysis, along with the previously recovered glacial-tectonic features and the presence of glacial sediments on the East Siberian continental shelf and slope, implicate that glacial ice not only grew out from the East Siberian shelf but also from the De Long Islands, and that there were also ice rafted sediments delivered to the southern Lomonosov Ridge from westerly sources, e.g., the eastern Laptev Sea.

**Acknowledgements**

We thank the supporting crew and Captain of I/B Oden and the support of the Swedish Polar Research Secretariat for realizing the cruise and coring operations. This research is part of the University of Oulu funded project Loss of Ice in the Arctic System (LIAS): geological perspective of global environmental change. This research was partially supported by the Finnish Cultural Foundation. Many thanks to Riitta Kontio from the Oulu Mining School Research Center for laboratory assistance. Special thanks to Leena Palmu and Pasi Juntunen from the Centre for Material Analysis (CMA) at the University of Oulu for assistance with FESEM. We thank Rüdiger Stein and Leonid Polyak for their constructive and useful comments on the manuscript.

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
