# Peer review of "Heavy mineral assemblages of the De Long Trough and southern Lomonosov Ridge glacigenic deposits: implications for the East Siberian Ice Sheet extent"

_Climate of the Past, 2021_

## Referee Comment (RC1)

Alatarvas, R., O'Regan, M., Strand, K.:
**The De Long Trough: defining the mineralogical signature of the East Siberian Ice Sheet**
(Manuscript submitted to "*Climate of the Past – Discussions*")

Review by Ruediger Stein (11 October 2021)

Whereas quite detailed information about the existence, timing and extent of the North American and Eurasian ice sheets during the Pleistocene glaciations is available, the data base for the existence, timing and extent of an East Siberian Ice Sheet is very limited and based on sparse geophysical and marine geological data (e.g., Jakobsson et al., 2010; Niessen et al., 2013; Jakobsson et al., 2014). In order to get more detailed information about the history of this ice sheet, Alatarvas et al. present new mineralogical data from marine sediment cores recovered from the East Siberian shelf and slope, especially from a glacial trough and related trough mouth fan setting, focusing on heavy mineral assemblages. That means, heavy minerals in the coarse fraction have been used to reconstruct provenance and source areas and plausible transport mechanisms of the terrigenous sediment fractions. These data are the basis for far field reconstructions of ice sheet activity on the east Siberian Margin during the late Quaternary. For lithostratigraphy and chronology of the studied sediment cores, an important and fundamental prerequisite for any paleo-reconstructions, the authors refer to O`Regan et al. (2017, 2020) as well as Jakobsson et al. (2016). This very well written paper is certainly of interest and an important puzzle piece for the still needed more detailed reconstruction of the history of the East Siberian Ice Sheet, especially in context and relationship to the other major circum-Arctic ice sheets. The new heavy-mineral data give evidence for an extensive ice sheet growth from the East Siberian shelf but also from the New Siberian Islands and westerly sources, probably during MIS 6 (although a deposition during a stadial in MIS 5 or the glacial period of MIS 4 might also be possible as stated by the authors). In general, I have a very positive opinion about the outcome of this study and would like to see the paper published. However, I have some points that should be considered before publication (see below). Thus, at its present stage I recommend "**publication after minor revision**".

Several Polarstern expeditions have been carried out in the area across and around southern Lomonosov Ridge close to the Siberian continental margin (e.g., Rachor, 1997; Stein, 2015, 2019), and a large number of sediment cores have been recovered (Fig. 1). Most of these sediment cores can be correlated very well based on their lithology, and a very clear lithostratigraphic concept has been developed (Fig. 1a) that is further supported by physical property data (see Marine Geology subchapters in the cruise reports Rachor, 1997; Stein, 2015, 2019). Based the lithostratigraphy and physical property records as well as some micropaleontological data and preliminary interpretation of paleomag data from Core PS2757-8, a tentative (!) age model had been proposed in our early studies (cf., Behrends, 1999; Stein et al., 2001), an age model that is still be used (cf., Stein et al., 2017) although it's still tentative. Based on this age model, the prominent dark gray sandy silty clay unit in the lower part of the cores seems to be of MIS 6 age (Fig.1a). The lithologies of the key cores of this study can also be correlated to the Polarstern cores, and their age model based on the new findings of O'Regan et al. (2020) seems to support the old tentative age model we have used for our Polarstern cores.

From several of these Polarstern cores (including key cores PS2757 and 2761) detailed mineralogical and geochemical data have been produced within three PhD studies (Behrends, 1999; Müller, 1999; Schoster, 2005; part of the data is published in Behrends et al., 1999; Müller and Stein, 2000; Schoster et al., 2000). These data including heavy minerals, clay minerals, and major & minor elements, have been used to reconstruct (1) the provenance, source areas and transport mechanisms of the terrigenous sediment fractions and, based on these data sets, (2) the history of the Eurasian and East Siberian ice sheets (Fig. 2). The extent and timing of proposed ice sheets in northern Siberian during MIS 4 and/or MIS 6 are discussed (Fig. 2b; cf., Arkhipov et al., 1986,1995; Müller, 1999). As one example, the heavy mineral record from Core PS2757 is shown in Figure 2c. I recommend that some of these data should be considered and discussed in the present paper.

Finally, I would like to highlight that the reconstruction of provenance, source areas and transport mechanisms of the terrigenous sediment fractions as well as the history of the Pliocene-Pleistocene Eurasian and East Siberian ice sheets is one of the key objectives of the IODP Expedition 377 (ArcticOcean Paleoceanography – ArcOP) scheduled for autumn 2022 (Stein et al., 2021). The locations of the potential IODP sites are in the neighbourhood of the cores discussed here (Fig. 1b). Thus, the results of the studies by Alatarvas et al. as well as our own previous studies on Polarstern material may give ground truth information that is important and helpful for the interpretation of the coming IODP data.

**References**

Arkhipov, S.A., Bespaly, V.G., Faustova, M.A., Glushkova, O., Isayeva, L.L.,Velichko, A.A., 1986. Ice-sheet reconstructions. Quaternary Science Reviews 5, 475-483.

Arkhipov, S.A., Ehlers, J., Johnson, R.G., Wright, H.E., 1995. Glacial drainage towards the Mediterranean during the Middle and Late Pleistocene. Boreas 24, 196-206.

Behrends, M., 1999. Reconstruction of sea-ice drift and terrigenous sediment supply in the Late Quternary:  Heavy-mineral associations in sediments of the Laptev-Sea continental margin and the central Arctic Ocean. Reports on Polar Research 310, 167 pp. (PhD Thesis University of Bremen; in German).  https://epic.awi.de/id/eprint/26490/1/BerPolarforsch1999310.pdf

Behrends, M., Hoops, E., Peregovich, B., 1999. Distribution patterns of heavy minerals in Siberian rivers, the Laptev Sea and the eastern Arctic Ocean: an approach to identify sources, transport and pathways of terrigenous matter. In: H. Kassens, *et al.* (Eds.), Land-Ocean Systems in the Siberian Arctic: Dynamics and History, Springer, Berlin (1999), pp. 265-286.

Jakobsson, M., Polyak, L., Edwards, M., Kleman, J., Coakley, B., 2008. Glacial geomorphology of the Central Arctic Ocean: the Chukchi Borderland and the Lomonosov Ridge. Earth Surf. Process. Landforms 33, 526–545.

Jakobsson, M., Andreassen, K., Bjarnadóttir, L. R., Dove, D., Dowdeswell, J. A., England, J. H., Funder, S., Hogan, K., Ingólfsson, Ó., Jennings, A., Larsen, N. K., Kirchner, N., Landvik, J. Y., Mayer, L., Mikkelsen, N., Möller, P., Niessen, F., Nilsson, J., O'Regan, M., Polyak, L., Nørgaard-Pedersen, N., and Stein, R. 2014. Arctic Ocean glacial history, Quaternary Sci. Rev. 92, 40–67, https://doi.org/10.1016/j.quascirev.2013.07.033 .

Jakobsson, M., Nilsson, J., Anderson, L., Backman, J., Björk, G., Cronin, T.M., Kirchner, N., Koshurnikov, A., Mayer, L., Noormets, R., O'Regan, M., Stranne, C., Ananiev, R., Barrientos Macho, N., Cherniykh, D., Coxall, H., Eriksson, B., Flodén, T., Gemery, L., Gustafsson, Ö., Jerram, K., Johansson, C., Khortov, A., Mohammad, R., Semiletov, I.: Evidence for an ice shelf covering the central Arctic Ocean during the penultimate glaciation. Nature Communications 7, 10365, https://doi.org/10.1038/ncomms10365

Jakobsson, M., Nilsson, J., O'Regan, M., Backman, J., Löwemark, L., Dowdeswell, J.A., Mayer, L., Polyak, L., Colleoni, F., Anderson, L.G., Björk, G., Darby, D., Eriksson, B., Hanslik, D., Hell, B., Marcussen, C., Sellén E., and Wallin, Å., 2010. An Arctic Ocean ice shelf during MIS 6 constrained by new geophysical and geological data, Quaternary Sci. Rev. 29, 3505–3517, https://doi.org/10.1016/j.quascirev.2010.03.015

Müller, C., 1999. Reconstruction of paleoenvironmental conditions at the Laptev Sea continental margin during the last two glacial/interglacial cycles based on sedimentological and mineralogical investigations. Reports on Polar 328, 146 pp. (PhD Thesis University of Bremen; in German). https://epic.awi.de/id/eprint/26507/1/BerPolarforsch1999328.pdf

Müller, C. and Stein, R., 2000. Variability of fluvial sediment supply to the Laptev Sea continental margin during Late Weichselian to Holocene times: Implications from clay-mineral records. Int. Journ. Earth Sci. 89, 592-604. https://link.springer.com/content/pdf/10.1007/s005310000112.pdf

Niessen, F., Hong, J. K., Hegewald, A., Matthiessen, J., Stein, R., Kim, H., Kim, S., Jensen, L., Jokat, W., Nam, S., 2013. Repeated Pleistocene glaciation of the East Siberian continental margin, Nature Geosci., 6, 842–846, https://doi.org/10.1038/ngeo1904,

O'Regan, M., Backman, J., Barrientos, N., Cronin, T.M., Gemery, L., Kirchner, N., Mayer, L.A., Nilsson, J., Noormets, R., Pearce, C., Semiletov, I., Stranne, C., Jakobsson, M., 2017. The De Long Trough: a newly discovered glacial trough on the East Siberian continental margin. Climate of the Past 13, 1269–1284, https://doi.org/10.5194/cp-13-1269-2017

O'Regan, M., Backman, J., Fornaciari, E., Jakobsson, M., West, G., 2020. Calcareous nannofossils anchor chronologies for Arctic Ocean sediments back to 500 ka. Geology, 48 (11), p. 1115–1119. http://doi.org/10.1130/G47479.1

Rachor, E (Ed.), 1997. Scientific cruise report of the Arctic Expedition ARK-XI/1 of RV „Polarstern" in 1995. Rep. Pol. Res. 226, 157 pp. https://epic.awi.de/id/eprint/26404/1/BerPolarforsch1997226.pdf

Schoster, F., 2005. Terrigenous sediment supply and paleoenvironment in the Arctic Ocean during the late Quaternary - Reconstructions from major and trace elements. Reports on Polar and Marine Research 498, 149 pp. (PhD Thesis University of Bremen; in German). https://epic.awi.de/id/eprint/26677/1/BerPolarforsch2005498.pdf

Schoster, F., Behrends, M., Müller, C., Stein, R., Wahsner, M., 2000. Modern river discharge in the Eurasian Arctic Ocean: Evidence from mineral assemblages and major and minor element distributions. Int. Journ. Earth Sci. 89, 486-495. https://link.springer.com/content/pdf/10.1007/s005310000120.pdf

Stein, R. (Ed.), 2015. The Expedition PS87 of the Research Vessel *Polarstern* to the Arctic Ocean in 2014. Reports on Polar and Marine Research 688, 273 pp (http://epic.awi.de/37728/1/BzPM_0688_2015.pdf).

Stein, R. (Ed.), 2019. The Expedition PS115/2 of the Research Vessel POLARSTERN to the Arctic Ocean in 2018. Reports on Polar and Marine Research 728, 249 p.; (https://epic.awi.de/id/eprint/49226/1/BzPM_0728_2019.pdf ).

Stein, R., Fahl, K., Gierz, P., Niessen, F., Lohmann, G., 2017. Arctic Ocean sea ice cover during the penultimate glacial and the last interglacial. Nature Communications 8: 373. https://www.nature.com/articles/s41467-017-00552-1

Stein, R., Fahl, K., and Müller, J., 2012. Proxy reconstruction of Arctic Ocean sea ice history: „From IRD to $IP_{25}$". Polarforschung 82, 37-71, https://epic.awi.de/id/eprint/31715/1/Polarforschung_82-1_37-71.pdf

Stein, R., Matthiessen, J., Niessen, F., Krylov, R., Nam, S., Bazhenova, E., 2010. Towards a Better (Litho-) Stratigraphy and Reconstruction of Quaternary Paleoenvironment in the Amerasian Basin (Arctic Ocean). Polarforschung 79 (2), 97-121. https://epic.awi.de/id/eprint/22435/1/Ste2010b.pdf

Stein, R., St. John, K., and Everest, J., 2021. Expedition 377 Scientific Prospectus: Arctic Ocean Paleoceanography (ArcOP). International Ocean Discovery Program. https://doi.org/10.14379/iodp.sp.377.2021

**Figure Caption**

**Fig. 1**
**(a)** Transect of selected sediment cores recovered across the southern Lomonosov Ridge during Polarstern expeditions in 1995, 2014, and 2018 showing main lithologies, lithostratigraphy, and still tentative age model (MIS 6 to 1) based on shipboard data and core correlation (Rachor, 1997; Stein, 2015, 2019 and further references therein). Core PS115/2-14-3 was recovered at the location of proposed IODP Site LR-06A. **(b)** Map shows locations of Polarstern cores (white circles: 1995; blue circles: 2014; black circles: 2018) and eight of the proposed ArcOP sites (large yellow circles). Location of Core 29-GC-1 shown as black rhomb. Figure from Stein et al. (2021), supplemented.

**Fig. 2**
**(a)** Bathymetric map of the Arctic Ocean modified to showing lowered sea level of 120 m during maximum glaciation and ice streams (blue and red arrows), projected flow lines of ice shelves and limits of ice rises (from Jakobsson et al., 2008, supplemented). Main source areas of specific minerals are shown (from Stein et al., 2010, 2012 and references therein): qua = quartz; dol = dolomite; ill = illite; sme = smectite; chl = chlorite; kao = kaolinite; am = amphibole; cli = clinopyroxene. Colour codes mark source region: green = western Laptev Sea, Kara Sea, Barents Sea; blue = eastern Laptev Sea, East Siberian Sea; orange = Bering Strait; pink = Canada, northern Greenland; white = no specific source area. Tentative extent of East Siberian Ice Sheet has been added (cf., Niessen et al., 2013). (Figure from Stein et al., 2012, supplemented). **(b)** Siberian shelves and proposed ice sheets in northern Siberia during MIS 4 and MIS 6. Figure from Müller (1999) based on Arkhipov et al. (1986, 1995) and own data from her PhD thesis work. Core locations are indicated by black circles. **(c)** Distribution of heavy minerals in Core PS2757. MIS boundaries 5/6, 4/5, 3/4, and 1/2 are marked as dashed intervals based on a preliminary age model (Behrends, 1999).

[Figure]

Fig. 1

[Figure]

(a)

(Figure from Stein et al., 2012, supplemented)

(b)

Core locations

Extent of
ice sheets
(MIS 4 and 6)

Water depth of shelf areas

von 50 m bis 100 m

von 0 m bis 50 m

(Figure from Müller 1999, based on Arkhipov et al. 1986 and 1995)

(c) Heavy mineral distribution in Core PS2757

(Behrends, 1999)

Fig. 2

---

## Author Comment (AC1)

**Response to RC1**

We appreciate Prof. Stein's valuable comments and suggestions, as we highly appreciate his proven experience and wide publication background with Arctic research. We are therefore grateful for the opportunity to improve the manuscript according to his comments and ideas. Comments seem very positive and are constructive, and also give some valid references to be considered. He said that manuscript is well written paper and is certainly of interest and an important puzzle piece for the still needed more detailed reconstruction of the history of the East Siberian Ice Sheet, especially in context and relationship to the other major circum-Arctic ice sheets.

**Response to specific comments:**

**Comment:** The lithologies of the key cores of this study can also be correlated to the Polarstern cores, and their age model based on the new findings of O'Regan et al. (2020) seems to support the old tentative age model we have used for our Polarstern cores. From several of these Polarstern cores (including key cores PS2757 and 2761) detailed mineralogical and geochemical data have been produced within three PhD studies (Behrends, 1999; Müller, 1999; Schoster, 2005; part of the data is published in Behrends et al., 1999; Müller and Stein, 2000; Schoster et al., 2000). These data including heavy minerals, clay minerals, and major & minor elements, have been used to reconstruct (1) the provenance, source areas and transport mechanisms of the terrigenous sediment fractions and, based on these data sets, (2) the history of the Eurasian and East Siberian ice sheets (Fig. 2). The extent and timing of proposed ice sheets in northern Siberian during MIS 4 and/or MIS 6 are discussed (Fig. 2b; cf., Arkhipov et al., 1986,1995; Müller, 1999). As one example, the heavy mineral record from Core PS2757 is shown in Figure 2c. I recommend that some of these data should be considered and discussed in the present paper.

**Response:** We will respond to that comment by adding an additional paragraph with highlighted references in the discussion where the results from the Polarstern cruises are discussed especially related to heavy mineral distributions. We will focus on some similarities with Behrand's (1999) work on PS2757, which could be quite similar to the data from SWR-29GC.

**Comment:** Stein highlighted that the reconstruction of provenance, source areas and transport mechanisms of the terrigenous sediment fractions as well as the history of the Pliocene-Pleistocene Eurasian and East Siberian ice sheets is one of the key objectives of the IODP Expedition 377 (ArcticOcean Paleoceanography – ArcOP) scheduled for autumn 2022. The locations of the potential IODP sites are in the neighborhood of the cores discussed here. Thus, the results of the studies by Alatarvas et al. as well as our own previous studies on Polarstern material may give ground truth information that is important and helpful for the interpretation of the coming IODP data.

**Response**: This is a relevant notice and importance of this type of studies and the upcoming ArcOP expedition will be mentioned in discussion of the manuscript.

---

## Author Comment (AC2)

**Response to RC2**

We appreciate Leonid Polyak's very detailed comments and suggestions on the manuscript. We are thankful for the chance to adjust the manuscript according to his comments and suggestions. Comments and suggested references are valid and constructive.

**Response to major comments:**

**Comment:** In particular, a lack of samples from the LR core above the MIS6 diamicton doesn't allow for a comparison with interglacial/deglacial environments and with the ESS cores, which represent different geographic, depositional, and possibly stratigraphic settings. Also, the ESS shelf record is poorly represented by just two samples from the last deglaciation.

**Response**: We now specifically concentrate on correlating diamicts between the studied cores and improve our interpretations. This is actually the primary aim of this study. Our sample set in 24-GCI is useful to do compositional comparison between diamicts and overlying sediments and not for detecting several transitions from glacial to interglacial.

**Comment:** In addition to the new data, the paper provides a compilation of potential source rocks, which is useful both for this and future provenance studies in the region. It would be more logical, however, to present this information in the Study Area section rather than in the Discussion.

**Response:** A compilation figure and table of the potential source rocks is now presented in the Study Area section.

**Comment:** The discussion is not well structured and can be difficult to follow, especially without a graphic summary.

**Response:** The discussion will be structured better, and a summary table or graphic summary added.

**Comment:** The discussion in section 5.3 is even more confusing as the interpretation of the new data is mixed with inferences from or attributed to prior studies, lumped together in one paragraph. At the same time, depositional environments and processes are not adequately explained

**Response:** Depositional environments and processes of diamicts will be explained as a separate paragraph within section 5.3. At the same time relevant references can be added.

**Comment:** Identifying the ice-sheet provenance from the data under study is problematic as sediment delivered by ice directly from the mainland may not be easily distinguishable from sediment redeposited from the shelf. This task is even more complicated by multiple mechanisms of glacigenic sedimentation, such as subglacial till, proglacial debris flows, icebergs, etc. As these processes are not identified or even discussed in the paper, I don't see how the authors can reconstruct the ice origin.

**Response:** Depositional environments and processes will be discussed more adequately as a separate paragraph including identification of different glacigenic processes for generating diamicts.

**Comment:** The inference on different sources for glacigenic sediments in cores from the DLT and LR is more convincing and informative. However, it raises questions too. Most important, the SW provenance of the MIS6 diamicton in the LR core is inconsistent with the direction of the eroding ice flow indicated by the seafloor bedforms (Jakobsson et al., 2016). One possibility is that the erosional event is not reflected in the sedimentary record (hiatus), while the diamicton was deposited from icebergs. In any case, this issue needs to be discussed.

**Response:** This is a good notice. The SW provenance of the MIS6 diamicton in the LR core and the direction of the eroding ice flow indicated by the seafloor bedform will be discussed. Regarding the 'SW sources', there are a couple of options here aside from different transport/re-deposition processes. In the paper by West et al., 2021 he shows that the base of 29-GC may not be the actual glacial diamict associated with scouring of the Ridge, but a later one, maybe break up and large-scale iceberg inputs. This opens the door for more SW sources of material being transported there. This will be added to the discussion.

**Comment:** References are limited. Only studies of heavy minerals are used for discussing the ESS sediments, while papers dealing with other mineralogical aspects could provide a more comprehensive context (e.g., Washner et al., 1999; Viscosi-Shirley et al., 2003; Nwaodua et al., 2014; Ye et al., 2020). Relevant studies of the distribution and composition of glacigenic deposits at or adjacent to the ESS margin are also missing (e.g., Schreck et al., 2018; Joe et al., 2020; Ye et al., 2020). A broader Arctic Ocean context can be derived from recent provenance papers (e.g., Dong et al., 2020; Xiao et al., 2021).

**Response:** References will be added for other mineral aspects supporting interpretations of heavy mineral distribution. The distribution and composition of glacigenic deposits at the ESS margin will be added compiled with the results from the Polarstern cruises.

**Response to additional comments:**

**Comment:** In Abstract; "This study concentrates on defining the mineralogical signature and dynamics of the ESIS (p. 1, lines 16-17)". This statement is misleading. The study deals with mineralogical signature of sediments from the ESS margin. Whether it reflects the ESIS provenance is a matter of interpretation, even more so for the ice-sheet dynamics.

**Response:** The study will mainly deal with mineralogical signature of sediments from the ESS margin.

**Comment:** In Introduction; "… previous studies have suggested the existence of ice sheets over parts of the East Siberian continental shelf during the larger Pleistocene glaciations following the mid-Pleistocene transition (Colleoni et al., 2016; Niessen et al., 2013), the Saalian (Marine Isotope Stage 6) (Jakobsson et al., 2016)" (p. 2, lines 2-4). This statement is inaccurate and confusing. Are we talking about multiple glaciations or just the MIS6? And where does the MPT come from? While there is evidence for a very large impact of the MIS6 glaciation on the Arctic, we do not know whether it featured the largest ice sheet on the East Siberian margin. Niessen et al. (2013) demonstrated glacial seafloor features in this region but have not constrained their age. Later studies suggested a very extensive glacial footprint in at least some parts of the East Siberian and Chukchi margin for MIS4 (Schreck et al., 2018; Joe et al., 2020; Kim et al., 2021).

**Response:** This statement will be revised for the suggested glacial extents.

**Comment:** In Materials and Methods; What is the point for a detailed description of seismostratigraphy? These data are not used in the paper.

**Response:** The description of seismotratigraphy will be shortened and condensed. Overall, description of seismostratigraphy can help correlation between the studied cores.

**Comment:** In Discussion; What are "our mineral assemblages" (p. 13, line 19)? Please be specific - What is "the eastern sector of the East Siberian Ice Sheet" (p. 13, line 27)? So far, the extent and configuration of this, largely hypothetical ice sheet is very poorly understood. What can be inferred from the data is that the mineralogical signature indicates delivery from the eastern part of the ESS.

Response: "These results can be detected also within our studied heavy mineral assemblages." It is accurate, that the mineralogical signature indicates only delivery from the eastern part of the ESS.

**Comment:** In Conclusions, "This suggests that due to dynamics of the ice flow and deposition the glacial ice not only grew out from the East Siberian shelf but also from the New Siberian Islands and westerly sources"

(p. 14, lines 10-11). How could the ice sheet grow from the new Siberian Islands, if it was advancing on the islands from the north (Nikolskiy et al., 2013)? And what are the "westerly sources"?

**Response:** This sentence could be revised, and the following references taken into count. "There could have been a smaller local ice cap developed over the De Long Islands during a stadial of MIS 5 (O'Regan et al. 2017)." "The ice stream occupying the DLT was likely connected to glacial ice over the De Long and New Siberian Islands (O'Regan et al. 2017)." Westerly sources relate mostly to the Laptev Sea.

**Comment:** In Terminology; I don't think the "Central plateau" is a good term for the study area as the entire East Siberian shelf is pretty flat. This term has been used indeed by Naugler et al., 1974, but it doesn't make much geomorphic or geological sense. More generic terms like "inner shelf" or just "shelf" would be more appropriate.

**Response:** More generic terms like "inner shelf" or just "shelf" can be seen appropriate.

**Recommendations:** Overall, I believe the MS requires a considerable revision. Ideally would be to investigate a few more samples to fill the gaps in the sedimentary record under study, notably from post-MIS6 sediments in the LR core and from the Holocene on the ESS shelf. However, I understand the practical constraints. The text, especially the Discussion, needs to be better articulated, with a clear delineation of inferences from the data reported and a more comprehensive and to the point use of information from prior studies. A summary figure would be very helpful for following and comprehending the interpretation. The conclusions, abstract, and the title need to be coherent with the data-based interpretation. An accurate title would be something like "Heavy mineral provenance of glacigenic deposits at the East Siberian margin, Arctic Ocean".

**Response for recommendations:** There is no specific need for few more samples as detecting several transitions from glacial to interglacial is not a target in this study. We specifically concentrate on correlating diamicts within each studied core and improve our interpretations for existing glacial processes and diamict provenances. The text can be articulated better, and more accurate title will be considered.

**Cited references:**

O'Regan, M., Backman, J., Barrientos, N., Cronin, T.M., Gemery, L., Kirchner, N., Mayer, L.A., Nilsson, J., Noormets, R., Pearce, C., Semiletov, I., Stranne, C., Jakobsson, M.: The De Long Trough: a newly discovered glacial trough on the East Siberian continental margin. Climate of the Past, 13, 1269–1284, https://doi.org/10.5194/cp-13-1269-2017, 2017.

West, G., Alexanderson, H., Jakobsson, M., and O'Regan, M. Optically stimulated luminescence dating supports pre-Eemian age for glacial ice on the Lomonosov Ridge off the East Siberian continental shelf. Quaternary Science Reviews, 267, https://doi.org/10.1016/j.quascirev.2021.107082, 2021.

---

## Author Response (AR1)

**A list of all relevant changes made in the manuscript:**

- Title has been changed: "Heavy mineral assemblages of the De Long Trough and southern Lomonosov Ridge glacigenic deposits: implications for the East Siberian Ice Sheet extent"
- Abstract has been revised
- Chapter 5.1 "Parent rocks for heavy minerals" has been moved to chapter 2. "Regional and geological setting" as 2.3 "Geological provinces and parent rocks for heavy minerals"
- Figure 6 has been moved from chapter 5 to 2 (now Fig. 2), and it has been modified by adding features for sediment transport, inferred ice flow and ice rafting
- Table 3 has been moved from chapter 5 to 2 (now Table 1)
- Chapter 3.1 has been revised and shortened. The title has changed to "Sedimentary and acoustic units", and descriptions have been compressed
- Figure 3 is figure 4 now and grain size of core 29-GC1 (West et al., 2021) has been added
- Discussion has been revised overall, clear delineation of inferences from the reported data and a more comprehensive and to the point use of information from prior studies has been carried out
- A synthesis figure of the heavy mineral analysis and interpretations of this study related to transportation and depositional sites has been added to the discussion (Fig. 7)
- The abstract and conclusions have been revised.
- New references have been added to the list
- Some words and a few sentences have been corrected, added, deleted, or changed throughout the manuscript

**Response to RC1**

We appreciate Prof. Stein's valuable comments and suggestions, as we highly appreciate his proven experience and wide publication background with Arctic research. We are therefore grateful for the opportunity to improve the manuscript according to his comments and ideas. Comments seem very positive and are constructive, and also give some valid references to be considered. He said that manuscript is well written paper and is certainly of interest and an important puzzle piece for the still needed more detailed reconstruction of the history of the East Siberian Ice Sheet, especially in context and relationship to the other major circum-Arctic ice sheets.

**Response to specific comments:**

**Comment:** Several Polarstern expeditions have been carried out in the area across and around southern Lomonosov Ridge close to the Siberian continental margin (e.g., Rachor, 1997; Stein, 2015, 2019), and a large number of sediment cores have been recovered (Fig. 1). Most of these sediment cores can be correlated very well based on their lithology, and a very clear lithostratigraphic concept has been developed (Fig. 1a) that is further supported by physical property data (see Marine Geology subchapters in the cruise reports Rachor, 1997; Stein, 2015, 2019). Based the lithostratigraphy and physical property records as well as some micropaleontological data and preliminary interpretation of paleomag data from Core PS2757-8, a tentative (!) age model had been proposed in our early studies (cf., Behrends, 1999; Stein et al., 2001), an age model that is still be used (cf., Stein et al., 2017) although it's still tentative. Based on this age model, the prominent dark gray sandy silty clay unit in the lower part of the cores seems to be of MIS 6 age (Fig.1a). The lithologies of the key cores of this study can also be correlated to the Polarstern cores, and their age model based on the new findings of O'Regan et al. (2020) seems to support the old tentative age model we have used for our Polarstern cores.

From several of these Polarstern cores (including key cores PS2757 and 2761) detailed mineralogical and geochemical data have been produced within three PhD studies (Behrends, 1999; Müller, 1999; Schoster, 2005; part of the data is published in Behrends et al., 1999; Müller and Stein, 2000; Schoster et al., 2000). These data including heavy minerals, clay minerals, and major & minor elements, have been used to reconstruct (1) the

provenance, source areas and transport mechanisms of the terrigenous sediment fractions and, based on these data sets, (2) the history of the Eurasian and East Siberian ice sheets (Fig. 2). The extent and timing of proposed ice sheets in northern Siberian during MIS 4 and/or MIS 6 are discussed (Fig. 2b; cf., Arkhipov et al., 1986,1995; Müller, 1999). As one example, the heavy mineral record from Core PS2757 is shown in Figure 2c. I recommend that some of these data should be considered and discussed in the present paper.

**Response:** We have mentioned the correlation of our studied key cores to many of the sediment cores previously recovered around the area during various Polarstern expeditions (e.g., Rachor, 1997; Stein, 2015, 2019) and to mineralogical and geochemical data produced from the cores (Behrends, 1999; Müller and Stein, 2000; Schoster et al., 2000). The suggested previous tentative age models of these cores have also been discussed (cf., Stein et al., 2017; O'Regan et al., 2020), and in addition, an age-depth model by West et al. (2021).

**Comment:** Finally, I would like to highlight that the reconstruction of provenance, source areas and transport mechanisms of the terrigenous sediment fractions as well as the history of the Pliocene-Pleistocene Eurasian and East Siberian ice sheets is one of the key objectives of the IODP Expedition 377 (ArcticOcean Paleoceanography – ArcOP) scheduled for autumn 2022 (Stein et al., 2021). The locations of the potential IODP sites are in the neighbourhood of the cores discussed here (Fig. 1b). Thus, the results of the studies by Alatarvas et al. as well as our own previous studies on Polarstern material may give ground truth information that is important and helpful for the interpretation of the coming IODP data.

**Response**:This is a relevant notice and importance of this type of studies and the upcoming ArcOP expedition is mentioned in the manuscript.

**Response to RC2**

We appreciate Leonid Polyak's very detailed comments and suggestions on the manuscript. We are thankful for the chance to adjust the manuscript according to his comments and suggestions. Comments and suggested references are valid and constructive.

**Response to major comments:**

**Comment:** In particular, a lack of samples from the LR core above the MIS6 diamicton doesn't allow for a comparison with interglacial/deglacial environments and with the ESS cores, which represent different geographic, depositional, and possibly stratigraphic settings. Also, the ESS shelf record is poorly represented by just two samples from the last deglaciation.

**Response**: We have now specifically concentrated on correlating diamicts between the studied cores and improve our interpretations. This actually was the primary aim of this study. Our sample set of the core 24-GCI is now utilized in compositional comparison between diamicts and overlying sediments and not to detect several transitions from glacial to interglacial.

**Comment:** In addition to the new data, the paper provides a compilation of potential source rocks, which is useful both for this and future provenance studies in the region. It would be more logical, however, to present this information in the Study Area section rather than in the Discussion.

**Response:** A compilation figure and table of the potential source rocks is now presented in the Study Area (Regional and geological setting) section.

**Comment:** The discussion is not well structured and can be difficult to follow, especially without a graphic summary.

**Response:** The hole discussion chapter is revised and structured better now, and a summary figure has been added.

**Comment:** The discussion in section 5.3 is even more confusing as the interpretation of the new data is mixed with inferences from or attributed to prior studies, lumped together in one paragraph. At the same time, depositional environments and processes are not adequately explained.

**Response:** The discussion has been revised; interpretation of the new data is separated from the consideration of previous studies. Depositional environments and processes are explained more.

**Comment:** Identifying the ice-sheet provenance from the data under study is problematic as sediment delivered by ice directly from the mainland may not be easily distinguishable from sediment redeposited from the shelf. This task is even more complicated by multiple mechanisms of glacigenic sedimentation, such as subglacial till, proglacial debris flows, icebergs, etc. As these processes are not identified or even discussed in the paper, I don't see how the authors can reconstruct the ice origin.

**Response:** Depositional environments and processes are discussed more adequately including identification of different glacigenic processes.

**Comment:** The inference on different sources for glacigenic sediments in cores from the DLT and LR is more convincing and informative. However, it raises questions too. Most important, the SW provenance of the MIS6 diamicton in the LR core is inconsistent with the direction of the eroding ice flow indicated by the seafloor bedforms (Jakobsson et al., 2016). One possibility is that the erosional event is not reflected in the sedimentary record (hiatus), while the diamicton was deposited from icebergs. In any case, this issue needs to be discussed.

**Response:** The SW provenance of the MIS 6 diamicton in the southern LR core is discussed. Paper by West et al. (2021) suggesting that the base of core 29-GC may not be the actual glacial diamict associated with scouring of the ridge, but possibly later occurred ice sheet rafting, and large-scale iceberg input is referred to in the discussion.

**Comment:** References are limited. Only studies of heavy minerals are used for discussing the ESS sediments, while papers dealing with other mineralogical aspects could provide a more comprehensive context (e.g., Washner et al., 1999; Viscosi-Shirley et al., 2003; Nwaodua et al., 2014; Ye et al., 2020). Relevant studies of the distribution and composition of glacigenic deposits at or adjacent to the ESS margin are also missing (e.g., Schreck et al., 2018; Joe et al., 2020; Ye et al., 2020). A broader Arctic Ocean context can be derived from recent provenance papers (e.g., Dong et al., 2020; Xiao et al., 2021).

**Response:** References of other mineral aspects have been added for supporting the interpretations. In addition, features for sediment transport, inferred ice flow and ice rafting have been discussed and added to the provenance area figure.

**Response to additional comments:**

**Comment:** In Abstract; "This study concentrates on defining the mineralogical signature and dynamics of the ESIS (p. 1, lines 16-17)". This statement is misleading. The study deals with mineralogical signature of sediments from the ESS margin. Whether it reflects the ESIS provenance is a matter of interpretation, even more so for the ice-sheet dynamics.

**Response:** The statement has been revised: **"This study concentrates on defining the heavy mineral signature of glacigenic deposits from the East Siberian continental margin from core collected during the 2014 SWERUS-C3 expedition".**

**Comment:** In Introduction; "… previous studies have suggested the existence of ice sheets over parts of the East Siberian continental shelf during the larger Pleistocene glaciations following the mid-Pleistocene transition (Colleoni et al., 2016; Niessen et al., 2013), the Saalian (Marine Isotope Stage 6) (Jakobsson et al., 2016)" (p. 2, lines 2-4). This statement is inaccurate and confusing. Are we talking about multiple glaciations or just the MIS6? And where does the MPT come from? While there is evidence for a very large impact of the MIS6 glaciation on the Arctic, we do not know whether it featured the largest ice sheet on the East Siberian margin. Niessen et al. (2013) demonstrated glacial seafloor features in this region but have not constrained

their age. Later studies suggested a very extensive glacial footprint in at least some parts of the East Siberian and Chukchi margin for MIS4 (Schreck et al., 2018; Joe et al., 2020; Kim et al., 2021).

**Response:** This statement has been replaced with "Seafloor mapping data now provide ample evidence for the existence of considerable ice masses on the East Siberian margin (Niessen et al., 2013; Jakobsson et al., 2014, 2016), but the timing and extent of these glaciations is still relatively unknown. According to West et al. (2021), there is a broad consensus on the lack of glacial activity on the Siberian shelf during the last glacial maximum (LGM), but when, and to what extent, the former ice sheets existed on the Siberian shelf remains poorly constrained by terrestrial evidence."

**Comment:** In Materials and Methods; What is the point for a detailed description of seismostratigraphy? These data are not used in the paper.

**Response:** Chapter 3.1 has been revised and shortened. The title has changed to "Sedimentary and acoustic units", and descriptions have been compressed. The information has been utilized in the correlation of the cores, and in the description of the deposits in the discussion.

**Comment:** In Discussion; What are "our mineral assemblages" (p. 13, line 19)? Please be specific - What is "the eastern sector of the East Siberian Ice Sheet" (p. 13, line 27)? So far, the extent and configuration of this, largely hypothetical ice sheet is very poorly understood. What can be inferred from the data is that the mineralogical signature indicates delivery from the eastern part of the ESS.

**Response:** These statements have been changed: "The mineral content variability of these rivers' sediments can be seen within the heavy mineral assemblages of the DLT diamict.", and "the eastern part of the ESS".

**Comment:** In Conclusions, "This suggests that due to dynamics of the ice flow and deposition the glacial ice not only grew out from the East Siberian shelf but also from the New Siberian Islands and westerly sources" (p. 14, lines 10-11). How could the ice sheet grow from the new Siberian Islands, if it was advancing on the islands from the north (Nikolskiy et al., 2013)? And what are the "westerly sources"?

**Response:** This statement has been revised: "The results from this heavy mineral analysis, along with the previously recovered glacial-tectonic features and the presence of glacial sediments on the East Siberian continental shelf and slope, implicate that glacial ice not only grew out from the East Siberian shelf but also from the De Long Islands, and that there were also ice rafted sediments delivered to the southern Lomonosov Ridge from westerly sources, e.g., the eastern Laptev Sea."

**Comment:** In Terminology; I don't think the "Central plateau" is a good term for the study area as the entire East Siberian shelf is pretty flat. This term has been used indeed by Naugler et al., 1974, but it doesn't make much geomorphic or geological sense. More generic terms like "inner shelf" or just "shelf" would be more appropriate.

**Response:** More generic terms like "inner shelf" or just "shelf" has now been used instead of central plateau.

**Recommendations:** Overall, I believe the MS requires a considerable revision. Ideally would be to investigate a few more samples to fill the gaps in the sedimentary record under study, notably from post-MIS6 sediments in the LR core and from the Holocene on the ESS shelf. However, I understand the practical constraints. The text, especially the Discussion, needs to be better articulated, with a clear delineation of inferences from the data reported and a more comprehensive and to the point use of information from prior studies. A summary figure would be very helpful for following and comprehending the interpretation. The conclusions, abstract, and the title need to be coherent with the data-based interpretation. An accurate title would be something like "Heavy mineral provenance of glacigenic deposits at the East Siberian margin, Arctic Ocean".

**Response for recommendations:** There is no specific need for few more samples as detecting several transitions from glacial to interglacial was not a target in this study. We specifically concentrated on correlating diamicts within each studied core and improved our interpretations for existing glacial processes and diamict provenances. Discussion has been revised overall, clear delineation of inferences from the reported data and a

more comprehensive and to the point use of information from prior studies has been carried out. A summary figure has been added. The abstract and conclusions have been revised. A new title "Heavy mineral assemblages of glacigenic deposits from the East Siberian continental margin: implications for the ice sheet extent".

**References:**

Jakobsson, M., Andreassen, K., Bjarnadóttir, L. R., Dove, D., Dowdeswell, J. A., England, J. H., Funder, S., Hogan, K., Ingólfsson, Ó., Jennings, A., Larsen, N. K., Kirchner, N., Landvik, J. Y., Mayer, L., Mikkelsen, N., Möller, P., Niessen, F., Nilsson, J., O'Regan, M., Polyak, L., Nørgaard-Pedersen, N., and Stein, R.: Arctic Ocean glacial history, Quaternary Science Reviews, 92, 40–67, https://doi.org/10.1016/j.quascirev.2013.07.033, 2014.

Jakobsson, M., Nilsson, J., Anderson, L., Backman, J., Björk, G., Cronin, T.M., Kirchner, N., Koshurnikov, A., Mayer, L., Noormets, R., O'Regan, M., Stranne, C., Ananiev, R., Barrientos Macho, N., Cherniykh, D., Coxall, H., Eriksson, B., Flodén, T., Gemery, L., Gustafsson, Ö., Jerram, K., Johansson, C., Khortov, A., Mohammad, R., Semiletov, I.: Evidence for an ice shelf covering the central Arctic Ocean during the penultimate glaciation, Nature Communications 7, 10365, https://doi.org/10.1038/ncomms10365, 2016.

Niessen, F., Hong, J. K., Hegewald, A., Matthiessen, J., Stein, R., Kim, H., Kim, S., Jensen, L., Jokat, W., and Nam, S.: Repeated Pleistocene glaciation of the East Siberian continental margin, Nature Geosci., 6, 842–846, https://doi.org/10.1038/ngeo1904, 2013.

West, G., Alexanderson, H., Jakobsson, M., and O'Regan.: Optically stimulated luminescence dating supports pre-Eemian age for glacial ice on the Lomonosov Ridge off the East Siberian continental shelf. Quaternary Science Reviews, 267, https://doi.org/10.1016/jquascirev.2021.107082, 2021.